# Energy comparison of sequential and integrated $CO_2$ capture and electrochemical conversion

Mengran Li [1], Erdem Irtem[1], Hugo-Pieter Iglesias van Montfort [1], Maryam Abdinejad [1] & Thomas Burdyny [1] ✉

Integrating carbon dioxide ($CO_2$) electrolysis with $CO_2$ capture provides exciting new opportunities for energy reductions by simultaneously removing the energy-demanding regeneration step in $CO_2$ capture and avoiding critical issues faced by $CO_2$ gas-fed electrolysers. However, understanding the potential energy advantages of an integrated process is not straightforward due to the interconnected processes which require knowledge of both capture and electrochemical conversion processes. Here, we identify the upper limits of the integrated process from an energy perspective by comparing the working principles and performance of integrated and sequential approaches. Our high-level energy analyses unveil that an integrated electrolyser must show similar performance to the gas-fed electrolyser to ensure an energy benefit of up to 44% versus the sequential route. However, such energy benefits diminish if future gas-fed electrolysers resolve the $CO_2$ utilisation issue and if an integrated electrolyser shows lower conversion efficiencies than the gas-fed system.

Carbon dioxide ($CO_2$) capture and subsequent conversion represents a promising route for the production of fossil-fuel-free fuels and feedstocks from waste $CO_2$. In the past two decades, these capture and conversion steps have separately been advanced through innovations that have led to continuously lower implementation costs and higher energy efficiencies for each process[1]. For example, $CO_2$ capture can be operated at an overall cost of US\$50–150 to capture one tonne of $CO_2$ using commercially mature amine scrubbing processes from industrial sources[2,3]. Capture processes also show the potential to operate using alkaline capture sorbents[4] at \$94–232 or solid sorbents[5,6] at about \$600 to capture one tonne of $CO_2$ from the air. On the conversion side, low-temperature $CO_2$ electrolysers using pure $CO_2$ feeds have achieved a current density beyond $1 \, A \, cm^{-2}$ to convert $CO_2$ selectively to feedstocks (e.g., carbon monoxide (CO) and ethylene ($C_2H_4$))[7–10]. However, $CO_2$ electrolysis requires efforts to better define its role with the upstream $CO_2$ capture and downstream separation processes and understand the impacts of the processes-

associated energy penalties (e.g. electrolyte recovery, product separation)[11,12].

Given the eventual need to combine $CO_2$ capture and electrochemical conversion processes, and the diminishing energy efficiency returns from optimizing each process separately, researchers have considered the techno-economic and energy benefits of integrating capture and conversion[13,14]. For chemical processes, the discussion of whether 'to integrate, or not to integrate' $CO_2$ capture processes with conversion has been proposed to reduce overall energy requirements. For example, in a techno-economic study, $CO_2$ capture and chemical conversion showed a potential energy advantage of up to 46% when integrated to produce chemicals such as methyl formate from hydrogen, $CO_2$, and methanol (where the methanol serving as the capture media) at high pressures[15,16]. With the potential for $CO_2$ electrolysis at room temperature to act as a means of $CO_2$ conversion, we ask a similar question: does an integrated $CO_2$ capture and conversion process offer potential energy or process advantages over a sequential

[1]Materials for Energy Conversion and Storage (MECS), Department of Chemical Engineering, the Delft University of Technology, van der Maasweg 9, 2629 HZ Delft, The Netherlands. ✉e-mail: t.e.burdyny@tudelft.nl

capture and electrochemical conversion process? Here an integrated approach implies that the electrochemical process converts the captured $CO_2$ (e.g., carbamate and bicarbonate) in a captured medium. In this work, we construct and compare these two scenarios to answer two key questions: (1) does an integrated route have energy advantages over the sequential route; (2) what performance metrics need to be met within the integrated electrolysis for such a route to be viable?

The scope of this work is limited to the $CO_2$ capture process based on commercially available monoethanolamine-based amine scrubbing techniques and the $CO_2$ electrochemical conversion to CO in gas-fed electrolysers and amine-based capture media. Shown in Fig. 1 are two comparable scenarios for a sequential capture and conversion process (Fig. 1a) and an envisioned integrated approach based on $CO_2$-to-CO in amine capture media (Fig. 1b). In the sequential route, the captured $CO_2$ is released at high purity via an amine-scrubbing step and then compressed and fed as a gas to a $CO_2$ electrolyser unit. Product separation and (bi)carbonate regeneration processes are included in the conversion step. The product is diluted due to the presence of unreacted $CO_2$ and needs to be separated from $CO_2$ through pressure-swing adsorption. In the $CO_2$ gas-fed electrolyser unit, $CO_2$ gas tends to form carbonate and bicarbonate ions (denoted as (bi)carbonates) by reacting with the hydroxide ions from electrochemical reduction (i.e., $CO_2$ reduction and hydrogen evolution reaction), as shown in reactions (1) – (5). Usually, only less than 50% of $CO_2$ gas molecules consumed in the electrolyser contribute to CO production[17–19]. The (bi)carbonates could either cross over the membrane[20] to the anolyte or precipitate at the cathode[21]. The (bi)carbonates in the electrolyte can be regenerated back to $CO_2$ gas and hydroxide anolyte by reacting calcium hydroxide to form calcium carbonate precipitates. The precipitates will then be calcinated to release $CO_2$ and produce calcium oxide that will be hydrated to become calcium hydroxide in the final step[4].

$$CO_2 + H_2O + 2e^- \Longleftrightarrow CO + 2OH^- \tag{1}$$

$$2H_2O + 2e^- \Longleftrightarrow H_2 + 2OH^- \tag{2}$$

$$CO_2(g) + 2OH^- \Longleftrightarrow CO_3^{-2} + H_2O \tag{3}$$

$$CO_2(g) + OH^- \Longleftrightarrow HCO_3^- \tag{4}$$

$$HCO_3^- + OH^- \Longleftrightarrow CO_3^{-2} + H_2O \tag{5}$$

In an integrated process, there is an opportunity for the $CO_2$ electrolyser to displace the stripper unit by converting the captured $CO_2$ while regenerating the capture medium simultaneously[22,23] (see Fig. 1b). In the amine-scrubbing cases, such a displacement could save 88–203 kJ $mol_{CO2}^{-1}$ from amine regeneration[24–29] and 14–19 kJ $mol_{CO2}^{-1}$ for compression[25,30], which accounts for up to 90% of the total energy consumption of the capture process[31]. In the integrated route, the $CO_2$-rich amines, containing substantially less free $CO_2$ (i.e., both $CO_2$ (g) and $CO_2$ (aq))[32,33], are directly fed into the integrated electrolysers. Although hydroxide ions are still produced from the electroreduction reactions, they will not form (bi)carbonate in the integrated electrolysis, because the majority of the captured $CO_2$ molecules are in the form of carbamate and bicarbonate. As such, the integrated electrolysis inherently avoids $CO_2$ gas loss faced by the gas-fed electrolysers[18,34–38]. It is important to note that the formation of bicarbonate in the $CO_2$ absorber (usually when $CO_2$ loading is >0.5 $mol_{CO2}$ $mol_{amine}^{-1}$) is not deemed as $CO_2$ loss, because it does not require a (bi)carbonate regeneration unit to recover $CO_2$. Therefore, there is no need for the integrated route to include a (bi)carbonate regeneration unit. If fulfilled, the integrated process may save >254 kJ $mol_{CO2}^{-1}$ to recover the $CO_2$ and hydroxide from the (bi)carbonates[4,18]. Finally, the sequential route requires an additional 14.5 – 36.4 kJ $mol_{CO2}^{-1}$ for product separation[39–41], which is avoided in the integrated case due to the spontaneous release of gas products from the capture

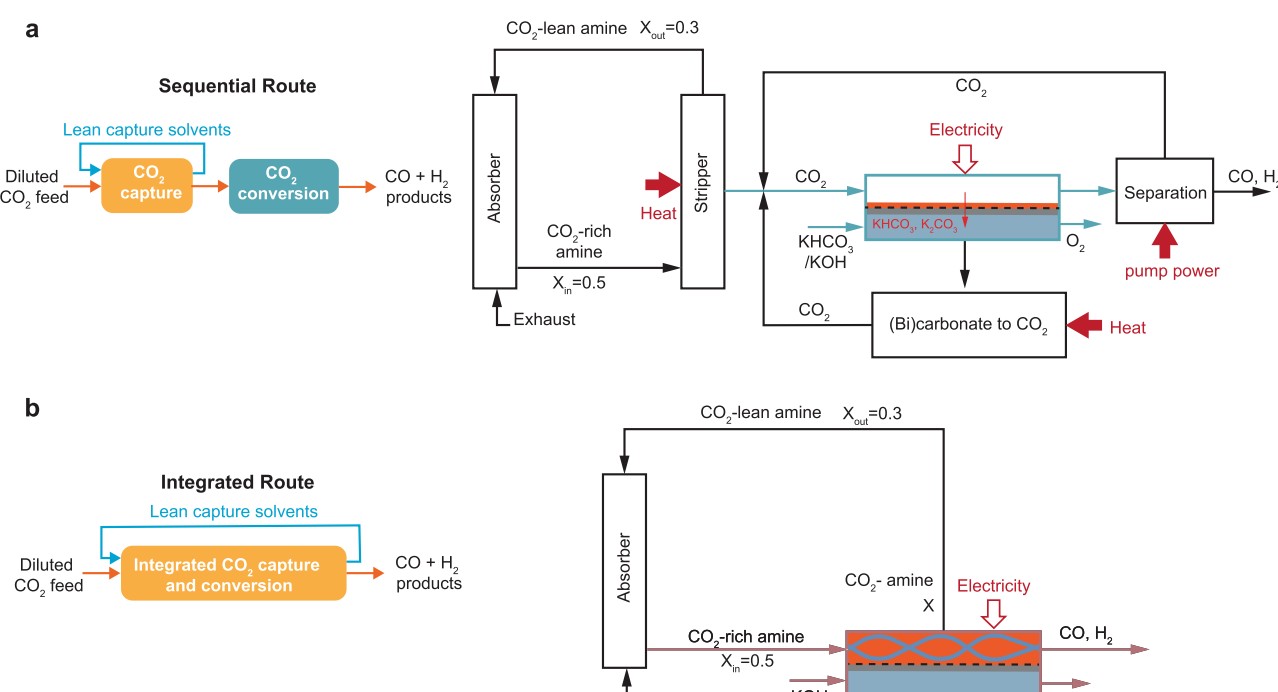

**Fig. 1 | Sequential and integrated routes of $CO_2$ capture and conversion.**
**a** Schematic illustration and block diagrams of the sequential route for amine-based $CO_2$ capture and electrolysis to produce CO. $CO_2$ electrolyser is based on membrane-electrode assemblies. **b** Schematic illustration and block flow diagrams of integrated $CO_2$ capture and direct $CO_2$ electroreduction from capture medium. The compression unit between stripper and electrolyzer is not shown in the block diagram. The $CO_2$ loadings of the $CO_2$-rich ($X_{in}$) and $CO_2$-lean amine ($X_{out}$) streams are assumed based on Gjernes et al. report[24].

media as a result of their low solubility (e.g., 1 mM for CO and 5 mM for $C_2H_4$ at 20 °C and 1 atm)[42]. Such a high-level analysis indicates that an ideal integrated route could save a total energy benefit of about 500 kJ $mol_{CO2}^{-1}$ when converting $CO_2$ to CO versus the sequential route.

However, there should be additional requirements for integrated electrolysis to be beneficial and replace amine regeneration and $CO_2$ compression in the sequential process. For instance, the integrated electrochemical conversion step needs to show at least similar performance metrics (cell voltage, Faradaic efficiency, and current densities) as the gas-fed electrolysers in the sequential process. Otherwise, energy gains for the overall process may be offset by the increased electrolyser energy requirements. Therefore, it is not straightforward to compare the energy benefits of an integrated process, thus warranting a more detailed analysis to help determine the upper limits of this new research direction.

Despite a number of reports on integrated electrolysis, their current performance is inferior to the gas-fed electrolysis system owing in part to their earlier development[22,43–47] (see Fig. 2) Regardless as the process can be evaluated as a function of performance metrics it is possible to forecast required performance targets at this early stage. Here, we compare the sequential and integrated scenarios from this high-level energy perspective, bringing in a wealth of current knowledge from both fields to give a perspective on the outlook of integrated $CO_2$ capture and electrochemical conversion. To perform this analysis, we compare the performance and working principles of the gas-fed and integrated electrolysers and discuss the essential roles of product Faradaic efficiency and cell voltages in the overall energy consumption to convert $CO_2$ into CO. Hydrogen is usually evolved as a side product together with $CO_2$ conversion, and CO product is mainly used together with hydrogen as the feedstock for downstream chemical manufacturing[48]. As such, this study is mainly focused on the energy required for $CO_2$ abatement over the $CO_2$ capture and conversion process. We then compare the sensitivity of various parameters to observe the parameter space where each process is favourable, which gives clear and targeted performance metrics for the novel integrated electrochemical conversion process. We finally conclude with an outlook on challenges and future potential for the integrated routes.

## Results

### Performance comparison for the gas-fed and integrated electrolysis

Here we compare the operation of existing gas-fed $CO_2$ electrolysers with future integrated electrolysers. We discuss the performance metrics for both conversion processes in-depth to provide a perspective on the comparative energy consumption of each route under different scenarios. We propose to gauge these two electrolyser types using the energy required to electrochemically convert one mole $CO_2$, which can be calculated from Eq. (6). The calculated energy is independent of the current densities, which allows us to compare these two electrolysers despite the levels of current densities achieved in prior literature.

As shown in Fig. 2a, the blue region highlights the energy requirement to produce CO with varied Faradaic efficiencies and cell voltages[49]. Overlayed within Fig. 2a are the existing state-of-the-art Faradaic efficiencies and current densities for the gas-fed electrolysis (blue circles). The integrated electrolysis (red circles) is relevant to the integrated route described in Fig. 1. For context, Fig. 2b communicates that product Faradaic efficiency has a more profound impact on energy consumption toward target CO than the cell voltages.

**Gas-fed $CO_2$ electrolysis to produce CO**. As a more advanced reaction, the gas-fed electrolyser outperforms the integrated electrolyser in product selectivity, current densities, and energy efficiency[22,43]. (see Fig. 2a) The state-of-the-art gas-fed $CO_2$ electrolysers can be operated at >100 mA $cm^{-2}$ with a cell voltage below 3–3.5 V and a product Faradaic efficiency of 80–90% (e.g., CO), as seen in Supplementary Table 3. When converting these performance metrics into the energy required to convert $CO_2$ into CO, we can estimate the benchmark gas-fed electrolyser to be in the range of 600–800 kJ $mol_{CO2-converted}^{-1}$. Our analysis uses only near-room-temperature flow cells and membrane-electrode assemblies (MEAs) for $CO_2$-to-CO as the model for the sequential route because this technology has a relatively high level of technical readiness[49–51].

In gas-fed $CO_2$ electroreduction, the dissolved $CO_2$ in water is the main catalytically reactant for the conversion, with $CO_2$ transported from a nearby gas phase[52,53]. High rates (up to 1 A $cm^{-2}$) are achieved by

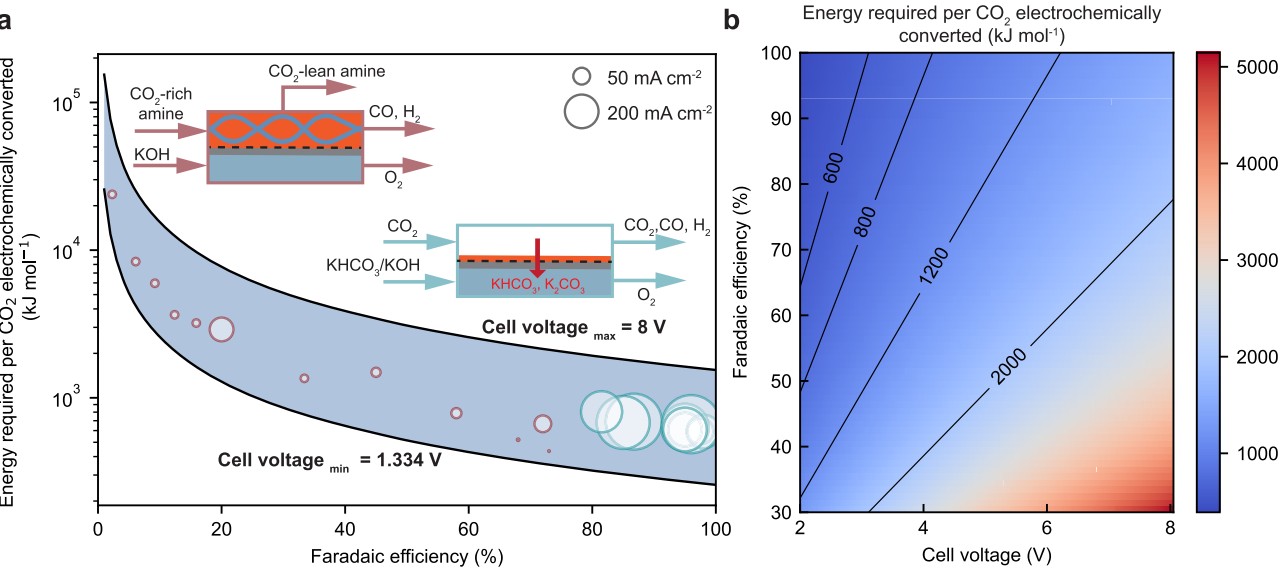

**Fig. 2 | Energy required to convert $CO_2$ to CO for gas-fed $CO_2$ electrolyser and direct $CO_2$ electrochemical upgrade from capture medium. a** The energy required to convert $CO_2$ to CO as a function of CO Faradaic efficiency with recently reported values for two different $CO_2$ electrolysers. Detailed data and references are summarized in Tables S2 and S3. The bubble size represents the magnitude of current densities for these cells as indicated in the legend. The insets illustrate the operating conditions of these two cells. **b** Impacts of CO Faradaic efficiency and cell voltages on the energy required of the $CO_2$ electrolysers. The solid lines indicate the Faradaic Efficiency vs. Cell voltage trends at certain energy requirements as indicated inline.

applying gas-diffusion electrodes[8–10], where the gases are transported from the gas channel to the catalyst facing the liquid electrolyte. Therefore, maintaining a stable electrode wettability is challenging for long-term operation[21].

In these gas-fed systems, the $CO_2$ utilisation efficiency is usually low (*e.g.*, capped at 50% if producing CO) due to carbonation between $CO_2$ and hydroxide ions (OH$^-$) at the cathode interface[18,34,35]. In an MEA configuration using an anion-exchange membrane, the (bi)carbonates migrated to the anode are reported to evolve back to $CO_2$[36–38]. Such $CO_2$ evolution should occur at the cost of increasing anode over-potentials by negatively affecting the anode reaction environment and the anode catalysts[54]. When the carbonate requires regeneration into $CO_2$, an energy penalty of at least 254 kJ $mol_{CO2\text{-converted}}^{-1}$ is associated with it in the case of 50% $CO_2$ utilisation efficiency. Our analysis then takes this energy penalty into account.

We acknowledge the recent efforts that attempt to remove the energy penalty associated with carbonate formation and low $CO_2$ utilisations, but these have not been demonstrated substantial overall performance metrics as compared to those presented in Fig. 2a. Such strategies use acidic environments and bipolar membranes to introduce protons to regenerate carbonate[17,55] or optimize local reaction environments or operating conditions[20,56]. For simplicity of this analysis, however, our analysis assumes a gas-fed $CO_2$ conversion of 50% with additional steps for product separation and carbonate regeneration processes.

**Electrolysis of the captured $CO_2$ in amine solutions**. In contrast to the gas-fed system above, reported electroreduction of captured $CO_2$ in monoethanolamine solutions presently has a higher energy requirement at low current densities (Fig. 2a and Supplementary Tables 1–2). The higher energy requirement of the integrated system is a result of the lower CO selectivity than the gas-fed systems. With further research efforts, these metrics are expected to improve.

In these systems, most cell potentials were unreported as it was not a primary part of the analysis. Hence, in order to populate Fig. 2a for the integrated case, we estimated the potentials associated with the anode, membrane, and electrolytes to perform a parameter sweep and evaluate the energy consumption for conversion (see Supplementary Note 1). Taking Lee et al.'s result as an example, the estimated cell voltage is 3 V to achieve 100 mA cm$^{-2}$ assuming the amine solution has the same ionic conductivity of 1 M KOH aqueous solution (21.5 S m$^{-1}$ for 1 M KOH solution[22,57]). The amine aqueous solution has a lower ionic conductivity than inorganic electrolyte (i.e. 3.7 S m$^{-1}$ for 5 M monoethanolamine solutions with about 0.4 $mol_{CO2}$ $mol_{amine}^{-1}$[58] as compared to 21.5 S m$^{-1}$ for 1 M KOH solution[57]). The ionic conductivity of the capture media can be effectively improved by including inorganic salts, such as $K_2SO_4$ and KCl[22,58]. As a result, the ohmic loss from the capture solvent can be significantly reduced, which is shown in Supplementary Fig. 1.

Further, The halide ions can serve as inhibitors to prevent oxidative degradation of amines[59,60], and the alkali cations are effective in promoting $CO_2$ electrochemical conversion[22,58,61]. Buvik et al.[60] also reported that the NaCl and KI salts show negligible impacts on the $CO_2$ capture capacity of the 30 wt% monoethanolamine solution. Nevertheless, further research efforts are needed to investigate the impacts of other inorganic salts on the properties of the capture media and the $CO_2$ absorption performance.

Due to the low CO Faradaic efficiency, the electrolysis of the existing early reports for captured $CO_2$ are at an energy consumption of 800 – 10$^4$ kJ $mol_{CO2}^{-1}$, as compared to the 600–800 kJ $mol_{CO2}^{-1}$ for the gas-fed system. From a state-of-the-art perspective, substantial energy reductions in the integrated electrolysis process are needed to make the overall integrated route more energetically favourable. The most straightforward path to reduce the energy load is through an increase in Faradaic efficiency for CO, which requires an

understanding of the underlying mechanisms and catalytically active species (e.g., carbamate ions, bicarbonate, or $CO_2$) dominating the conversion process. The outlook at the end of this article provides a detailed discussion of the mechanism for electrochemical $CO_2$ conversion and its challenges. To continue the analysis in Fig. 3, we put aside the performance metrics achieved in existing integrated reports and instead use three performance cases to see the energy comparison versus the sequential route.

**Determination of dominant energy contributors**
With the conversion processes described for the sequential and integrated routes, we can compare the expected energy requirement for both routes shown in Fig. 1 through a mass and energy balance. A detailed description of the models can be found in Supplementary Note 2, Supplementary Fig. 2, Supplementary Fig. 3 and Supplementary Table 4.

Here Fig. 3 explores the potential energy advantages of the integrated route under optimistic, baseline, and pessimistic performance metric scenarios for the electrolysis processes. Detailed conditions for these scenarios are summarized in Table 1 using the two most critical parameters for the integrated electrolysis process: CO Faradaic efficiency and cell voltage. The sequential route cases assume the gas-fed electrolyser to be operated at 3 V, 90% CO Faradaic efficiency, and 50% single-pass conversion. The 50%-$CO_2$-utilisation case assumes 50% of the reacted $CO_2$ convert to (bi)carbonate, while the 100% case assumes all the reacted $CO_2$ convert to CO molecules. It is important to note that the current density is not considered in the energy analysis, because current density predetermines the size and capital expense of the electrolysers, which is outside the scope of this work.

Our baseline condition is based on Lee et al.'s report that the Ag-coated ePTFE electrode can achieve 72% CO Faradaic efficiency at −0.8 V vs. reversible hydrogen electrode in monoethanolamine aqueous solutions. We believe the current densities can be further improved if applying hydrophilic 3D porous flow-through electrodes, as very recently reported by Zhang et al.[52] for the application of direct bicarbonate electroreduction. In the optimistic case, we anticipate the integrated electrolyser can perform similarly to the current gas-fed electrolyser. The pessimistic scenario assumes the future integrated electrolyser can only achieve a 40% CO Faradaic efficiency at a relatively large cell potential. All these three electrolysers are assumed to regenerate the capture media to a $CO_2$ loading at 0.3 $mol_{CO2}$ $mol_{amine}^{-1}$. We compared the sequential and integrated routes in terms of total energy, thermal energy and electricity, and energy cost.

In the sequential route, the energy consumption is shown to be dominated by $CO_2$ electrochemical conversion to produce CO, which includes $CO_2$ electrolysis (643 kJ $mol_{CO2}^{-1}$) and (bi)carbonate regeneration (254 kJ $mol_{CO2}^{-1}$). The $CO_2$ capture requires amine regeneration energy (179 kJ $mol_{CO2}$s$^{-1}$), $CO_2$ compression after capture (17 kJ $mol_{CO2}^{-1}$), and product purification (51 kJ $mol_{CO2}^{-1}$). These are all in terms of the amount of converted $CO_2$. Here the primary energy for the $CO_2$ electrolysis, compression, and product purification (based on pressure-swing adsorption) is electric work, but for (bi)carbonate and amine regeneration it is mainly inputted heat. The gas-fed $CO_2$ electrolyser was assumed to operate at a cell voltage of 3 V and a CO FE of 90%, which has been demonstrated experimentally (Fig. 2a).

When comparing the sequential route to the baseline integrated route, there is no foreseen overall energy advantage between the two routes (Fig. 3). The primary reason is the high energy requirement to convert $CO_2$, which offsets any foreseen energy benefits from process intensification. Considering the higher cost of electricity than heat, the integrated route in the baseline is in fact inferior to the sequential route due to its high electrical energy consumption (see Fig. 3b, c).

In the optimistic case, we assume the electrolysis of the captured $CO_2$ performs the same as the gas-fed electrolysis. In this scenario, the

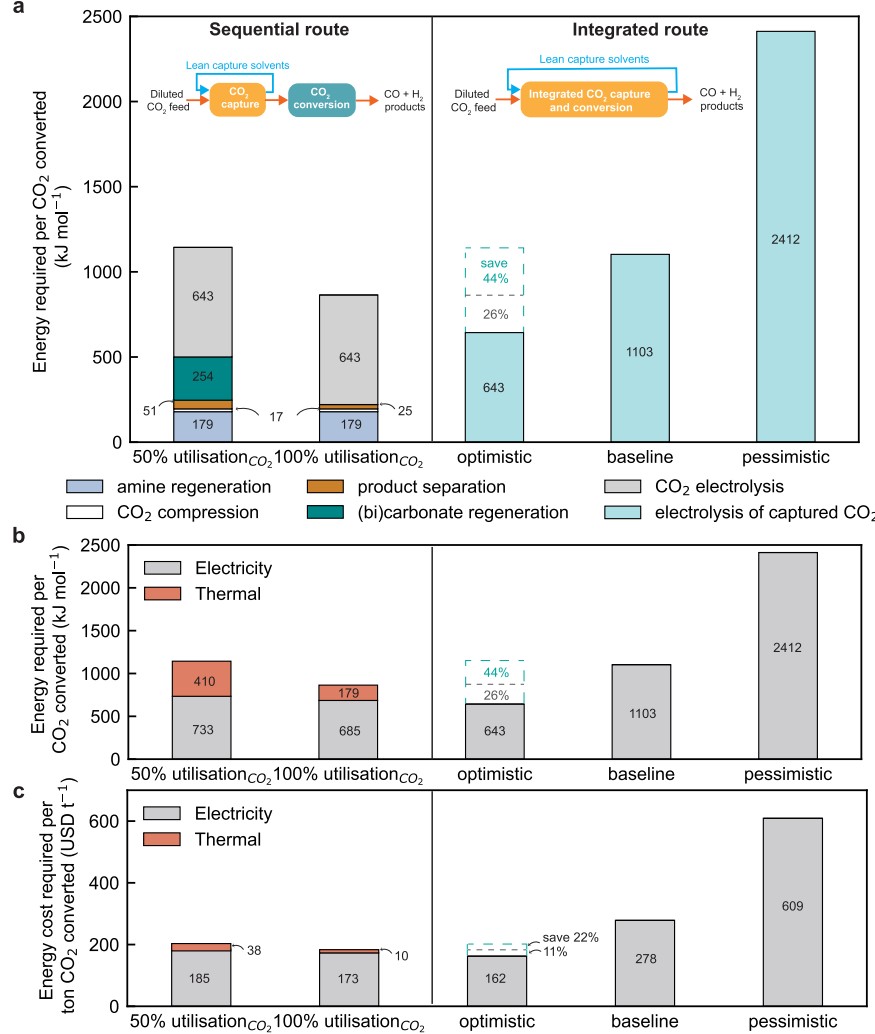

**Fig. 3 | Energy comparison between sequential and integrated routes in different scenarios.** Scenario analysis of (**a**) overall energy consumption, (**b**) thermal energy and electricity consumption, and (**c**) energy cost for sequential and integrated routes. In the sequential route, the $CO_2$ electrolyser includes state-of-the-art gas-fed electrolysers that show 50% $CO_2$ utilisation or future scenarios with 100% $CO_2$ utilisation. The optimistic, baseline and pessimistic electrolysis cases for the integrated routes are compared against the sequential route.

integrated route can save up to 44% of total energy due to a low cell voltage, high CO Faradaic efficiency, and no thermal energy associated with regeneration of amines (179 kJ $mol_{CO_2}^{-1}$) and (bi)carbonate (254 kJ $mol_{CO_2}^{-1}$), and electricity associated with $CO_2$ compression (17 kJ $mol_{CO_2}^{-1}$), and product purification (51 kJ $mol_{CO_2}^{-1}$). (Fig. 3b) The integrated route could save 22% energy cost over the sequential route. Such reduction in energy consumption renders the integrated route a more attractive option. Our results suggest most future research emphasis is placed on enhancing the Faradaic efficiency and cell voltages at industrially applicable current densities in order to reduce the

energy of the overall process. Without these conditions, the sequential route remains favourable.

In the pessimistic case, if the integrated route has a poor CO Faradaic efficiency (40%) and large cell voltage (5 V), however, the energy to drive integrated conversion is far higher (2412 kJ $mol_{CO_2}^{-1}$) than the gas-fed electrolyser, diminishing all the energy benefits from the process intensification. This scenario emphasises the importance of maximizing the two noted performance metrics.

Lastly, we assessed the energy consumption of the sequential route based on future $CO_2$ gas-fed electrolysis with 100% $CO_2$ utilization efficiency, meaning that no $CO_2$ gas will be lost into (bi)carbonate during gas-fed conversion. Very recent reports demonstrated the potential to improve $CO_2$ utilisation efficiency[53] by developing catalyst-membrane interface[44,54], optimising cell operating conditions (e.g., reducing $CO_2$ flow rates, increasing current densities, and optimising anolyte compositions and ionic strength)[46], or supplying protons towards the cathode to regenerate $CO_2$ from the (bi)carbonates, e.g., flowing strong acidic catholyte[22,55], applying cation-exchange membranes[44] or bipolar membrane[54] in a reverse mode. The single-pass conversion rate remains 50% in this optimistic sequential model, meaning that 50% of the inputted $CO_2$ feed converts to CO product and reduces the required pressure-swing absorption separation energy consumption. The total energy of such a sequential route is

**Table 1 | Summary of CO Faradaic efficiency and cell voltages for the integrated electrolyser in different scenarios**

| Scenarios | CO FE (%) | Cell voltage (V) |
| --- | --- | --- |
| Optimistic | 90 | 3 |
| Baseline | 70 | 4 |
| Pessimistic | 40 | 5 |

In all three models, the $CO_2$ specific energy requirement is assumed to be 179 kJ $mol_{CO_2}^{-1}$ for amine regeneration, 16.5 kJ $mol_{CO_2}^{-1}$ for $CO_2$ compression, 51 kJ $mol_{CO_2}^{-1}$ PSA product separation, and 254 kJ $mol_{CO_2}^{-1}$ (including 231 kJ $mol_{CO_2}^{-1}$ heat duty and 23 kJ $mol_{CO_2}^{-1}$ electricity) for (bi)carbonate regeneration. All these values are based on reported literature as listed in Supplementary note 2.

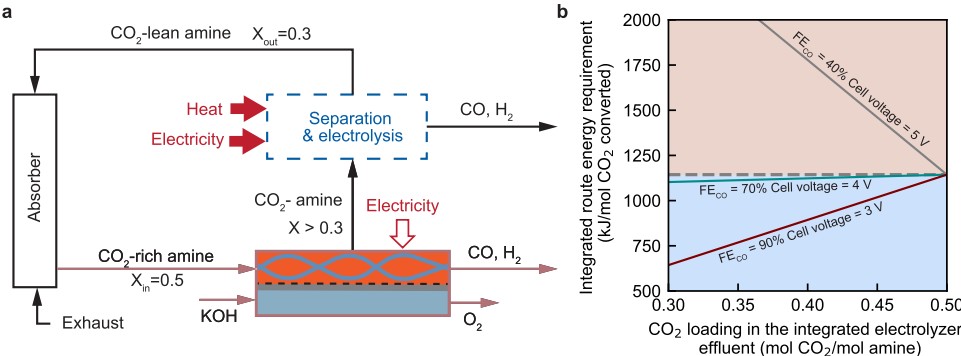

**Fig. 4 | Effect of the single-pass conversion of the integrated electrolyser on the overall energy efficiency. a** A schematic illustration of the integrated route where the electrolyser is unable to recover the capture media to the lean loading state. The separation and electrolysis process is symbolic process highlighted with a dashed box to regenerate the capture medium to the lean loading state. X represents the $CO_2$ loading in the capture medium, with a unit of mol $CO_2$ per mol amine molecule. **b** The energy comparison of the integrated route based on baseline (green solid line), pessimistic (grey), and optimistic (red) integrated electrolyser as a function of the electrolyser single-pass conversion. The grey dashed line represents the energy consumption of the sequential route based on state-of-the-art gas-fed $CO_2$ electrolysers. The blue region means that the integrated route is more energy-efficient than the sequential route, while the orange region indicates vice-versa.

864 kJ mol$^{-1}_{CO2}$ (see Fig. 3). Here the integrated optimistic case then only maintains a maximum overall energy advantage of 26% and energy cost benefit of 11%. We then conclude that if the energy penalty associated with (bi)carbonate regeneration is solved, there would be substantially lower energy gain possible by integrating capture and conversion even in the most optimistic scenario as described in this article.

Overall, our comparison highlights that energy benefits brought by the integrated route strongly depend on the progress in enhancing the energy efficiencies of the $CO_2$ electrolysis process. This trend makes sense because the $CO_2$ electrochemical conversion is the dominant contributor to the overall energy consumption, which is the primary reason preventing straightforward $CO_2$ capture and utilisation at a low cost.

### Single-pass conversion efficiency for the integrated electrolysis

In the analyses above, we assumed that the integrated electrolyser could recover the capture media to a lean loading state where it is directly recycled to the absorber (see Fig. 1b). If the electrolyser is unable to achieve the proposed lean state of 0.3 mol$_{CO2}$ mol$_{amine}^{-1}$, the high $CO_2$ loading (X > 0.3) in the lean amine stream will decrease the $CO_2$ absorption rate in the absorber unit. To maintain the overall $CO_2$ capture and conversion capacity of the process, adjustments to the process in Fig. 1b would then be needed. Here we discuss two possibilities, both of which will incur either additional capital or energy costs for the process.

One possible adjustment to account for lower conversions in the integrated electrolyzer is to increase the size of the absorber unit (Supplementary Fig. 4). A smaller difference between the low and high $CO_2$ loading states will then be present and a larger absorber allows for the same $CO_2$ capture capacity. Previous reports analysing the impacts on absorber size and capture costs of higher lean loading states indicate that an increase in lean loading from 0.3 to >0.4 mol$_{CO2}$ mol$_{amine}^{-1}$ would require 20–38% more capture costs[62,63]. With the electrolyser unit dominating the energy costs, however, these increased capture costs would be less substantial when considering the complete process. This option is also at a high technology readiness level.

A second option to maintain $CO_2$ capture and conversion capacity would be to add a secondary step after the integrated electrolyzer, which is a smaller version of the stripper and gas-fed electrolyzer unit from the sequential process (Fig. 4a). The energy implications of this option have yet to be explored in literature and will be examined within this section. In essence, this analysis examines the role of the single-pass conversion of the integrated electrolyser.

In the model, we included a symbolic process (including amine regeneration, gas-fed $CO_2$ electrolysis, product separation, and (bi)carbonate regeneration, shown in Fig. 4a) to regenerate the capture medium to the lean loading state and convert the rest of captured $CO_2$ to CO. In this case, the captured $CO_2$ in the effluent stream of the integrated electrolyser needs to be recovered to pure $CO_2$ gas from the regeneration unit and then fed into the gas-fed electrolyser for conversion.

We find that the role of the single-pass conversion efficiency is highly dependent on the performance of the integrated electrolyser. When the electrolyser operates at the baseline conditions, the capability of the integrated electrolyser to regenerate the capture medium becomes insignificant to the energy advantage of the integrated route. In contrast, if the electrolyser operates under either optimistic or pessimistic conditions, the single-pass conversion is essential for the overall energy consumption of the integrated route. The overall energy will benefit from an efficient electrolyser with high single-pass conversion. In contrast, a poorly performing electrolyser causes a significant overall energy penalty by increasing the single-pass conversion. This observation arises from the dominant role of the electrolysis in the overall energy of the capture and conversion process.

### Parameter sweeps of the integrated route

Here we briefly highlight how varied performance metrics of Faradaic efficiency and cell voltage impact the overall energy requirements for the integrated route. This analysis assumes the electrolyser can recover the capture medium to the lean loading state. Such an analysis provides a deeper context than the described optimistic, baseline and pessimistic scenarios above. Supplementary Fig. 5a shows that the energy advantage from the integrated route plummets linearly with the energy consumption of the integrated electrolysis. This trend highlights the core role of the electrolyser in determining the overall energy efficiency. The breakeven point for the integrated route is at the energy consumption of 1143 kJ mol$^{-1}$ for the integrated electrolyser (see Supplementary Fig. 5a). The value of the breakeven point should vary with the energy efficiency of the gas-fed electrolyser and the operating conditions, such as the single-pass conversion of the gas-fed electrolyser, energies to regenerate (bi)carbonate, amines, and to separate $CO_2$ and product. (see Supplementary Fig. 6).

The role of CO Faradaic efficiency and cell voltages were examined individually in influencing the energy gain from an integrated route. Supplementary Fig. 5b shows the breakeven point for CO Faradaic efficiency with varied cell voltages: the breakeven Faradaic efficiency is 51% at 3 V, 67% at 4 V, and 84% at 5 V. The impact from the

**Fig. 5 | Speciation of amine-based capture media in sequential and integrated routes and their impacts on $CO_2$ electrochemical conversion. a** Proposed integrated $CO_2$ absorption and electrolysis routes in amine-based solvents. **b** Schematic illustration of the role of alkali cations which promote interfacial charge transfer from the catalyst surface to the carbamate ions.

Faradaic efficiency is more significant than from the cell voltages, as shown in Supplementary Fig. 5b, c. The energy advantage from the integrated route decreases linearly with an increase of cell voltages and diminishes at 4.1 V when the Faradaic efficiency is 70%. Similarly, the breakeven cell voltages increase if the CO Faradaic efficiency could be further enhanced. Our analysis result indicates that the integrated $CO_2$ conversion as reported by Lee et al.[22], as shown in Fig. 2a, has the potential to achieve a more energy-efficient integrated route. Our model did not consider the cost associated with the current densities, which predetermine the capital cost of the electrolysers. Like the gas-fed $CO_2$ electrolysers, we believe operating at more than 200 mA cm$^{-2}$ with a high product selectivity is a prerequisite for an industrially relevant integrated system[64].

## Outlook for future integrated electrolysis

Our results identified that the electrochemical $CO_2$ conversion is the primary energy contributor for both sequential and integrated $CO_2$ capture and electrochemical conversion process. The reported energy efficiency of the integrated electrolyser is generally lower than the gas-fed $CO_2$ electrolysis. Such limitation originates from (1) the low surface coverage of reactants at the catalyst surface at industrially relevant rates and (2) the limited number of active sites the medium can reach over the hydrophobic gas-diffusion electrodes[22]. Therefore, the following research questions should be answered to advance the integrated electrolysers.

**What are the primary catalytically active species?.** It has been reported recently that the catalysts for gas-fed $CO_2$ electroreduction are selective to reduce $CO_2$ captured by amine-based capture media ($RNH_2$)[22,43,44,58]. In the $CO_2$-rich amines, the zwitterions ions including $RNHCO_2^-$ and $RNH_3^+$ are the major $CO_2$ species in the case of 30 wt% monoethanolamine aqueous solution when the $CO_2$ loading is below 0.4–0.6 mol $CO_2$ per mol amine[32,33]. Further increase of $CO_2$ loadings could promote carbamate hydrolysis to produce (bi)carbonates. Therefore, the $CO_2$ associated species should include carbamate ions, (bi)carbonate ions, and minor free dissolved $CO_2$, all may contribute to the $CO_2$ conversion.

However, there are still debates on the primary catalytically active species for the conversion in the amine (particularly for monoethanolamine) solutions. (see Fig. 5a) An early report by Chen et al.[43] claims that the free $CO_2$ dissolved in water can be the primary active species for the conversion, with nearly 100% Faradaic efficiency of hydrogen evolution regardless of the carbamate concentrations. In contrast, recent reports argued the possibility to reduce the carbamate ions as the main active reagent[22,61]. The claimed mechanisms for the direct carbamate reduction are different from the reduction mechanisms in $CO_2$ electrolysis[52] and direct bicarbonate reduction[65,66]. Interestingly, these recent reports also show an improvement of $CO_2$ conversion selectivity by increasing operating temperatures[22,45], which help release free $CO_2$. Therefore, the primary catalytically reactant for $CO_2$ conversion still remains a mystery but is paramount for the rational development of an efficient electrochemical system for integration.

In the $CO_2$ capture step based on 30 wt% monoethanolamine solutions, the $CO_2$ loadings are usually at 0.3–0.5 mole $CO_2$ per mole amine, meaning that the concentrations of the (bi)carbonate and free $CO_2$ are negligible. If the free $CO_2$ is the primary active reagent, regenerating and concentrating free $CO_2$ from carbamate and bicarbonate should be the key step to improving the integrated $CO_2$ conversion. Meanwhile, this strategy could adversely impact $CO_2$ capture. If the carbamate ions are the primary catalytically active species, they could be repelled by the negatively charged cathode surface, which might limit the coverage of reactants, especially at high overpotentials. Additionally, the active species need to diffuse to the negatively charged electrode through a thick hydrodynamic boundary layer usually >40 μm, if the integrated reactor configuration is similar to a $CO_2$-fed aqueous H-cell electrolyser (see Supplementary Fig. 7 for a comparison of aqueous versus gas-fed mass transport in $CO_2$ electrolysis)[67,68]. Efforts to improve integrated conversion at elevated current densities should then take such transport into consideration when designing such systems.

The results of our energy analysis indicate that the capture media for the integrated route could be designed to favour $CO_2$ conversion at a reasonable cost on $CO_2$ absorption. Therefore, an interdisciplinary collaboration between $CO_2$ capture and electrolysis is highly important to advance the integrated route.

**What are the pathways for the regeneration of the capture media?.** Complex homogenous equilibrium reactions often take place in the $CO_2$-capture medium system. In the sequential route, heating is required to drive the reactions towards the recovery of capture media and $CO_2$. Whereas the integrated route, as shown in Fig. 1b, uses electrochemical reactions to regenerate the capture medium via reduction of absorbed $CO_2$ and chemical-induced equilibria shift to the original states of the capture medium (see an example in Fig. 5a). Therefore, understanding the reaction equilibria under $CO_2$ electroreduction conditions is vital to the identification of chemical pathways to recover capture media inside the integrated electrolyser.

Similar to the gas-fed $CO_2$ electroreduction, hydroxide ions should also be produced at the catalyst surface as a by-product of water reduction and increase the pH locally around the electrode[69]. A prior report[70] has shown that the addition of a strong base (e.g., sodium hydroxide) to the $CO_2$-amine system could result in the formation of free amines and carbonate at the end equivalent points. As such, we could anticipate the formation of carbonate ions close to the electrode surface from the reactions between the hydroxide ions and unreacted $CO_2$ species. These carbonate ions could either reverse back to carbamate, free $CO_2$, or bicarbonate by reacting with the protons from the membrane[70,71] or stay as carbonate if additional cations are introduced into the cathode channel. The latter situation may cause operational issues for the integrated route such as inefficient $CO_2$ conversion, alteration of solvent chemistry, and potential carbonate salt precipitation from the solvent. Hence a dedicated control and balance of ions within the electrolyser also become critical in achieving an efficient amine recovery when using electrochemical $CO_2$ reduction as a regeneration step.

**How to improve integrated electrolyser performance?.** Including alkali cations such as potassium ions ($K^+$) or caesium ions ($Cs^+$) in the amine capture medium has shown its potential to improve $CO_2$ conversion efficiency[22]. As proposed by Lee et al.[22], the carbamate reduction can occur through an interfacial charge transfer mechanism, where the alkali cations can be packed (instead of protonated amines) at the electrode surface and facilitate charge transfer from the electrode surface to the carbamate ions, as illustrated in Fig. 5b. Meanwhile, an increasing number of reports also highlighted the essential role of alkali cations in activating gas-fed $CO_2$ electrochemical conversion[36,72,73]. Hence, the cations could synergistically minimise the surface coverage of protonated amines and activate $CO_2$ electroreduction. Nevertheless, the electrochemical reduction of the captured $CO_2$ is still low in CO Faradaic efficiency at >200 mA cm$^{-2}$, which could also be partially related to the limited electrochemical area due to the use of planar metal electrodes[43] or the hydrophobic nature of the gas-diffusion electrodes that are frequently used for gas-fed electrolysis[22].

We anticipate a significant improvement in $CO_2$ conversion rates (>200 mA cm$^{-2}$) by implementing new electrode structures such as hydrophilic 3D structured flow-through electrodes and optimised capture media[74,75]. The required diffusion distances of active species to achieve industrially applicable current densities are highly dependent on the concentrations and diffusion coefficients of the active species[76]. Therefore, understanding the primary active species and tailoring the local reaction environment could be effective in enhancing the $CO_2$ conversion rate in the integrated electrolysers.

Further, the desired wetting condition for the $CO_2$ conversion should have maximized solid-liquid interfaces with a minimal contact area of the gas bubble with the electrode surface. This means that the electrode surface should be hydrophilic, which is different from the desired wettability of gas-diffusion electrodes. Using metallic porous flow through electrodes is expected to achieve a high rate of $CO_2$ conversion by maximizing the electrochemical surface area, reducing the thickness of the boundary layer, and accelerating the detachment of gas products. On the other hand, more experimental and theoretical efforts are also essential to understand the potential catalyst surface restructuring, local reaction environment (e.g., pH and local concentration of amine species), and multiphase and ion transports in the cells, which have been demonstrated important for the stability and efficiency of the gas-fed $CO_2$ electrolysis[77–79].

## Discussion

Lastly, a directly coupled $CO_2$ capture and electrochemical conversion could potentially save close to 44% energy consumption and 21% energy cost versus a sequential process based on the state-of-the-art gas-fed $CO_2$ electrolysers, if the integrated electrolysis performs similarly to the gas-fed electrolysis (3 V and 90% CO Faraday efficiency) and has a high single-pass conversion efficiency to achieve the $CO_2$-lean state of the amines. However, this energy benefit drops from 44% to only 26% if new gas-fed $CO_2$ electrolysers with no $CO_2$ loss to (bi)carbonates emerged at high current densities. Our sensitivity analysis results suggest that research efforts should target an overall energy consumption at least similar to the gas-fed electrolyser performance for the integrated conversion cells so that the operational cost would not diminish the capital cost reduction from the process intensification. Although this work is a case study on the coupled amine scrubbing and $CO_2$-to-CO electrochemical conversion, our simple approach is anticipated to help researchers quickly understand the upper energy limits and targeted performance metrics for different integrated $CO_2$ capture and electrolysis processes. Collectively we hope this work provides a benchmark for integrated electrolysis research and provides a perspective to researchers and funding bodies seeking to achieve low-cost carbon capture and electrochemical utilisation processes through process integration.

## Methods

### Estimation of the specific energy required to electrochemically convert $CO_2$

We calculated the $CO_2$-specific energy requirement to produce CO from:

$$Energy\ required\ to\ convert\ 1\ mol\ CO_2 = \frac{E_{cell} \times j \times z \times F}{FE_{CO} \times j} \quad (6)$$

where $E_{cell}$ stands for cell voltage, $j$ for current density, $FE$ for Faradaic efficiency, $F$ for Faraday constant, $z$ for the number of charges to convert one $CO_2$ molecule ($z = 2$ for CO product).

We estimated the cell voltage by combining the estimated potential from anode, cathode, and ohmic loss. We did not consider the contribution from the Nernstian overpotential. We assumed the anode reaction for both gas-fed and integrated electrolysers is water oxidation from 1 M KOH basic solutions and described the reaction by using the Tafel equation. The potentials of cathode and ohmic loss from catholyte ($CO_2$-amine system with and without salts), anolyte, and membrane were calculated based on literature data. More details are discussed in the Supplementary Note 1.

$$E_{cell} = E_{anode} - E_{cathode} + \eta_{ohm} \quad (7)$$

### Mole and energy balances over the sequential and integrated routes

For simplicity of the analyses, the models only include the primary energy contributors, including the stripper, $CO_2$ compression, electrolysis, (bi)carbonate regeneration, and product separation. We conducted the mole balances of $CO_2$ and carbon over the electrolyser and calculated the $CO_2$-normalised energy requirement for each unit operations. The models are assumed to be in steady state with no losses of $CO_2$ or products. Most of the data are sourced from the literature data, which is summarised in Supplementary Note 2.

In our assumptions, more specifically, the amine scrubbing process used 30 wt% (or 5 M) monoethanolamine aqueous solution as the capture medium, with a $CO_2$-rich loading of 0.5 mol$_{CO2}$ mol$_{amine}^{-1}$ and a lean loading of 0.3 mol$_{CO2}$ mol$_{amine}^{-1}$ at the baseline conditions. The heat duty to recover $CO_2$ and amines is assumed to be 179 kJ mol$_{CO2}^{-1}$ at the baseline to reflect a higher technical readiness level for proof of concept. The compression of $CO_2$ is assumed to be 16.5 kJ mol$_{CO2}^{-1}$.

The baseline gas-fed $CO_2$ electrolyser has a 50% single-pass conversion and a 50% $CO_2$ utilisation efficiency, meaning that the 25% of the inputted $CO_2$ is converted to CO and the other 25% is converted to (bi)carbonate. In the optimistic case, the gas-fed electrolyser is assumed to be operated at 100% $CO_2$ utilisation, so no (bi)carbonate regeneration unit was included in this case. The recovery of $CO_2$ from the (bi)carbonate is estimated to be 254 kJ mol$^{-1}$, including 231 kJ mol$_{CO2}^{-1}$ heat duty and 23 kJ mol$^{-1}$ electricity[4]. The only products from the electrolysers are hydrogen and CO. The pressure swing adsorption serves to purify the product and recover the $CO_2$ for further conversion.

The baseline integrated electrolyser is assumed to recover the $CO_2$-lean amine and convert the absorbed $CO_2$ to CO. We assumed no downstream product separation is required for the integrated route. In the case where the recovery is incomplete, our analyses are focused on the additional energy requirement from the incorporation of additional process to recover and convert $CO_2$ as shown in Fig. 4a. We assumed the electricity is from low-carbon or zero-carbon power generation at a price of US$ 0.04 per kWh[23], and the heat duty from natural gas at a price of US$2.69 per million British Thermal Units[80]. At last, we performed the parameter sweep analyses to identify the effects of the key parameters on the overall energy requirement of the

sequential and integrated processes[81]. The detailed calculations and data are summarized in Supplementary Note 2.

## Data availability

The data generated in this study have been deposited in https://doi.org/10.5281/zenodo.7025326.

## Code availability

The code for the model and figures is available from GitHub under https://doi.org/10.5281/zenodo.7025326.

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

## Acknowledgements

T.B. and M.L. would like to acknowledge the European Union's Horizon 2020 research and innovation program under grant agreement No.

85144 (SELECT-CO2). T.B. would also like to acknowledge the NWO for an individual Veni grant.

## Author contributions

M.L. and T.B. wrote the manuscript. M.L. built the models. M.L., E.I., T.B., H.M., and M.A. performed the data analyses and visualisation. T.B. supervised the study. All authors contributed to the manuscript.

## Competing interests

The authors declare no competing interests.
