## [Peer Review File · Nature Communications]

Energy comparison of sequential and integrated CO₂ capture and electrochemical conversionReviewers' comments:

Reviewer #1 (Remarks to the Author):

This paper presents an energy analysis comparing conventional carbon capture and CO₂ electrochemical conversion to CO via sequential and then an integrated process. The authors examine several use cases assuming certain operating variables of the capture and electrochemical processes, which is boiled down into scenarios ranging from least to most optimistic. In the most optimistic case, the authors conclude that the integrated process can achieve up to 44% energy savings compared to the decoupled process; this expectedly drops somewhat if the integrated system cannot achieve good Faradaic and energy efficiencies. I find the most valuable finding to be the clear indication that the electrolysis dominates the energy costs, and therefore the energy savings from integration can range from zero (in fact, net harmful) to marginal to significant depending on how the electrolysis cell performs. Aside from this point, I find that there are a great many assumptions and even speculative aspects of the paper that lead me to assess low impact and rigor. Some of the most important of these are detailed further below. Given the number of serious concerns as well as the fact that the study appears to rest on a great many assumptions for a technology that is exceedingly early-stage, I do not find it suitable for publication in Nature Communications.

1. I would challenge a number of statements in the introduction that the authors take for granted. They assert that CO₂ capture and conversion "is a necessary step to mitigate anthropogenic climate change." This is not true according to many climate and policy experts who have made clear that carbon capture and sequestration are what is important; conversion to products that ultimately release CO₂ back to the environment upon use sooner or later raises a serious degree of doubt about what is necessary and impactful here. See for example <https://doi.org/10.1038/s41586-019-1681-6>, or Ref. 7 in the manuscript, which makes clear that the chemicals that can be made through utilization without adding more CO₂ to the atmosphere are those with small market size and rather limited mitigation potential.
2. Later, "To prove viable, however, conversion efforts now require integration with upstream CO₂ capture and downstream processes..." This is a strongly worded statement but is again a postulate, and not clear that it is true as the science is still nascent. In fact investigating this assumption is the objective of the paper. This is but one example where claims like this, which are important and carry weight, are made rather offhand.
3. The authors cite \$94-232 per ton CO₂ prices for DAC as having been broadly achieved (Ref 4), but this number is based on assumptions and small-scale operations. The price of DAC is highly controversial in literature, so the authors should acknowledge that this price is not established but estimated. Some numbers put DAC as high as \$1000/ton.
4. A major methodology concern is that the authors seem to assume that the amine needs to be fully electrochemically "regenerated" to very low loadings to be looped back to the absorber unit, otherwise necessitating an additional separation (desorption) process. This is not the case as many capture systems run with a small delta between rich and lean amine streams, e.g., 0.3-0.4 (see Y.-J. Lin, G.T. Rochelle / Chemical Engineering Journal 283 (2016) 1033-1043) while operating at steady state. There are good reasons not to fully regenerate and also not to run the amines to full loadings in the absorber. This may significantly alter the study's calculations and conclusions.
5. Table 1 describes three scenarios and suggests a coupling between CO FE and other parameters like mol CO₂ per mol amine (the parameter X). It is presented in such a way that it implies that high FE corresponds to lowest X, and vice versa. One could have full electrochemical regeneration of amine-CO₂ (thus low X) and make entirely parasitic products, thus have very low FE, or alternatively very small currents and large X, but very high FE. This calls into question the ensuing analysis.
6. Fig. 1 – the authors seem to have neglected pressurization needs between CO₂ capture and conversion in the sequential route.
7. The statement on p. 3, line 67 that the absence of CO₂ gas can avoid (bi)carbonate formation is not true. Many amines hydrolyze in the presence of water to yield bicarbonate – sometimes extensively more so than carbamate/ammonium, depending on the amine structure

(primary/secondary vs. tertiary and sterics), pKa, time to react, etc.

8. Overall, a major issue for this reviewer is that integrated capture and conversion is in very early stages, and as such, the tone of such an analysis, as well as delineation of how many assumptions are made and their limitations, is important. The tone of this work is more challenging and doubtful in places than is warranted given that the number of studies on this very nascent topic can be counted on one hand. I feel the work lacks scientific neutrality, by raising controversy where there is none. There is an assumption that all capture-conversion systems would, for instance, operate exactly like the one described in Lee et al., Nature Energy 2021. The authors must clearly indicate the limitations of their analysis and conclusions given how narrowly the capture-conversion technology has truly been considered in the work, and also that one or two studies should not be assumed to provide a firm foundation for performance metrics and strong debate about viability at this stage.

9. As a general comment, the methodology, as well as various details, are generally hard to follow in the main text. For instance, Fig. 2 can only be understood by reading the text, and lacks adequate labels, references, color guides (for b), etc. Throughout the paper, energy numbers used in the model are introduced in separate paragraphs or sections, which makes it hard to follow the calculations and independently confirm the findings.

Reviewer #2 (Remarks to the Author):

This work presents integrated CO₂ capture and electrochemical CO₂ conversion processes and compares it with conventional sequential processes. The topic of the study is appropriate for the journal because the amine-captured CO₂ reduction has great potential to eliminate energy-intensive CO₂ stripping processes and product separation processes. However, I do not support the publication of this manuscript in Nature Communications. First of all, this study only focuses on the energy analysis rather than comprehensive techno-economic analysis. As numbers of studies indicate, electrochemical processes include expensive unit operations such as electrolyzer, thus the capital investment cost cannot be overlooked. Authors emphasize the high FE throughout the paper but current density (related to size of electrolyzer) can be also a critical issue for the practical application.

I also do not agree with the carbonate stripper in both sequential and integrated routes in figure 1. Normally lean amine loading for the MEA CO₂ capture system ranges between 0.2-0.3, thus further CO₂ stripping may exaggerate the energy requirement. Also, process details of the systems are not presented in this work, thus I cannot evaluate the credibility of the model. Although the author did comprehensive analysis for the energy consumption but I think different types of energy cannot be compared using the same criteria. (e.g., thermal and electricity should be compared in terms of cost, not amount). Most importantly, carbamate reduction for the electrolysis of captured CO₂ electrolysis is very questionable. Numbers of studies indicate a strong C-N bond in the carbamate hinders direct conversion of CO₂, and addition of potassium ions or cesium ions may result in additional capital and operational costs due to their recovery and recycle. However, authors did not discuss the process of supplying cations and their recycling.

Detail comments are listed below

1. English Correction: first paragraph on page 2. Below are the parts found in paragraph 1.

- CO₂ capture & conversion "is"...
- CO₂ capture can (be) operate(d)...
- ... processes are also now operating (now being operated??)
- Low-temperature CO₂ electrolysis using pure CO₂ feeds "have" (not has)...

2. In page 5

"In molecular CO₂ reduction, dissolved CO₂ is the main catalytically reactant for the conversion" needs to be rephrased.

3. Page 7 Table 1: I wonder why the value for current density is not mentioned. Also, I'm curious how to choose a scenario for X. Both pessimistic and optimistic cases show lower X than baseline and it is very confusing. Corresponding results in Fig 3 b: the base case energy penalty for regeneration is the

highest in the base case. Please address proper logic behind of it In addition, in the middle paragraph of page 6, referring to Ref 17 (Nature Energy in 2021) as an example of the electrochemical reactor, it was 3.7V, 100 mA/cm², FE 20% (CO). but the baseline FE is set at 70%. I'd like to know the rationale too.

4. Eq.2-4 on page 8 of SI: The formula does not make sense. $F_{in}X$ is the amount of (X) converted in CO₂(F_{in}) entering the electrochemical reactor, which should be $F_{in} - F_p$, where $F_{in} - F_s = 2F_p$ in the formula. Perhaps the definition of conversion means electrochemical reduction & permeation through MEA. It would be good to express clearly in the body and SI.

5. SI page 9 Eq. 2-10 Equation following: Equation number is missing. Also, authors said $Q_c = Q_{sc}$ because $F_c = F_o$, but since both Q_c and Q_{sc} have units of kJ/mol, it does not seem to be related to F , the flow rate.

6. SI page 9 Eq.2-12: It said that PSA is a major energy item, but Q_{psa} (Eq. 2-10) of the PSA process is not included.

7 SI page 11 I recommend recheck carbon balance equation for example Eq. 2-16 seems $F_{ps} = F_a * X - F_a * X_{out} = F_a(X - X_{out})$. Why author divide ($X_{in} - X_{out}$)? To be sure, it is strongly recommended that authors disclose code/excel files etc.

Response to Reviewers' Comments

Reviewer #1

This paper presents an energy analysis comparing conventional carbon capture and CO₂ electrochemical conversion to CO via sequential and then an integrated process. The authors examine several use cases assuming certain operating variables of the capture and electrochemical processes, which is boiled down into scenarios ranging from least to most optimistic. In the most optimistic case, the authors conclude that the integrated process can achieve up to 44% energy savings compared to the decoupled process; this expectedly drops somewhat if the integrated system cannot achieve good Faradaic and energy efficiencies. I find the most valuable finding to be the clear indication that the electrolysis dominates the energy costs, and therefore the energy savings from integration can range from zero (in fact, net harmful) to marginal to significant depending on how the electrolysis cell performs. Aside from this point, I find that there are a great many assumptions and even speculative aspects of the paper that lead me to assess low impact and rigor. Some of the most important of these are detailed further below. Given the number of serious concerns as well as the fact that the study appears to rest on a great many assumptions for a technology that is exceedingly early-stage, I do not find it suitable for publication in Nature Communications.

Response:

We would like to express our gratitude to the Reviewers for their immense time and efforts in reviewing our manuscript which is clear from the detailed comments below. The summary and specific comments below have allowed us to constructively modify our manuscript such that our key points and novelties are clearer.

Importantly, we have further justified any parameter assumptions with further references and from the reviewer's comments to increase the scientific rigour. Quantitatively our conclusions remain the same after revision of key parameters such as the CO₂ loadings. Detailed point-by-point responses to the specific comments are below.

1. I would challenge a number of statements in the introduction that the authors take for granted. They assert that CO₂ capture and conversion "is a necessary step to mitigate anthropogenic climate change." This is not true according to many climate and policy experts who have made clear that carbon capture and sequestration are what is important; conversion to products that ultimately release CO₂ back to the environment upon use sooner or later raises a serious degree of doubt about what is necessary and impactful here. See for example <https://doi.org/10.1038/s41586-019-1681-6>, or Ref. 7 in the manuscript, which makes clear that the chemicals that can be made through utilization without adding more CO₂ to the atmosphere are those with small market size and rather limited mitigation potential.

Response:

We agree with the reviewer. We have corrected this statement, see Page 1 and below:

Carbon dioxide (CO₂) capture and subsequent conversion represents a promising route for the production of fossil-fuel-free fuels and feedstocks from waste CO₂.

2. Later, "To prove viable, however, conversion efforts now require integration with upstream CO₂ capture and downstream processes..." This is a strongly worded statement but is again a postulate, and not clear that it is true as the science is still nascent. In fact investigating this assumption is the objective of the paper. This is but one example where claims like this, which are important and carry weight, are made rather offhand.

Response:

We agree that the original phrasing was too strongly worded. What we were intending to say is that assessing CO₂ electrolysis as technology also requires consideration of its integration with intended upstream and downstream processes. We have provided the following revision on Page 1:

However, CO₂ electrolysis requires efforts to better define its role with the upstream CO₂ capture and downstream separation processes and understand the impacts of ...

3. The authors cite \$94-232 per ton CO₂ prices for DAC as having been broadly achieved (Ref 4), but this number is based on assumptions and small-scale operations. The price of DAC is highly controversial in literature, so the authors should acknowledge that this price is not establish but estimated. Some numbers put DAC as high as \$1000/ton.

Response:

We have revised the sentence to indicate the price uncertainty.

Capture processes also show the optimistic potential to be operated using alkaline capture sorbents⁴ at \$94 – 232 ...

4. A major methodology concern is that the authors seem to assume that the amine needs to be fully electrochemically “regenerated” to very low loadings to be looped back to the absorber unit, otherwise necessitating an additional separation (desorption) process. This is not the case as many capture systems run with a small delta between rich and lean amine streams, e.g., 0.3-0.4 (see Y.-J. Lin, G.T. Rochelle / Chemical Engineering Journal 283 (2016) 1033s–1043) while operating at steady state. There are good reasons not to fully regenerate and also not to run the amines to full loadings in the absorber. This may significantly alter the study’s calculations and conclusions

Response:

We appreciate the input and the correction, our apologies for considering the wrong CO₂-lean and -rich loadings in our modelling. We have now corrected this error by using a CO₂-rich amine loading of 0.5, and a CO₂-lean amine loading of 0.3 in our calculations and throughout the main text and supporting information. In addition, we have clarified that the term “fully regenerate” means to recover the capture media back to CO₂-lean loadings rather than completely regenerate to capture media with no CO₂ loaded.

From a process perspective, additionally, we have also separated the analysis in our text for greater clarity and an emphasis on the integrated electrolyser’s cell voltage and Faradaic efficiency. Specifically, the majority of our analysis is now based on assumptions that an integrated electrolyzer will be able to reach the CO₂ loading of 0.3 in its effluent CO₂-lean amine. We also added an additional discussion regarding the single-pass conversion and its implications in a separate section with results shown in Fig. 4.

However, such revision did not significantly change our conclusions from the models, meaning that this error plays a negligible role in evaluating the targeted energy advantage of the integrated route over the sequential route. After revision, for example, the upper limit of the energy advantage for the process integration changes from 44% to 42%. The loading of the CO₂-lean stream still serves an important role in affecting the energy advantages depending upon the electrolyser performance. The primary reason for such an inert response of the model to the error is the dominant role of the electrolyser in determining the overall energy efficiency.

Detailed revisions on Fig. 1, Fig. 3, Fig. 4, and Table 1 and the main texts are as below:

On page 3:

Fig. 1 Sequential and integrated routes of CO₂ capture and conversion. **a**, Schematic illustration and block diagrams of the sequential route for amine-based CO₂ capture and electrolysis to produce CO. CO₂ electrolyser is based on membrane-electrode assemblies. **b**, Schematic illustration and block flow diagrams of integrated CO₂ capture and direct CO₂ electroreduction from capture medium. The compression unit between stripper and electrolyzer is not shown in the block diagram. The CO₂ loading of the CO₂-rich and CO₂-lean amine streams are assumed based on Gjernes et al. report¹⁷.

On Pages 7-10

Determination of dominant energy contributors

With the conversion processes described for the sequential and integrated routes, we can compare the expected energy requirement for both routes shown in Fig. 1 through a mass and energy balance. Detailed description of the models can be found in Supplementary Note 2.

Here Fig. 2 explores the potential energy advantages of the integrated route under optimistic, baseline, and pessimistic performance metric scenarios for the electrolysis processes. Detailed conditions for these scenarios are summarized in Table 1 using the two most critical parameters for the integrated electrolysis process: CO Faradaic efficiency and cell voltage. The sequential route cases assume the gas-fed electrolyser to be operated at 3 V, 90% CO Faraday efficiency, and 50% single pass conversion. The with-bi-carbonate case assumes 50% of the reacted CO₂ convert to bicarbonate, while the without-bi-carbonate case assumes all the reacted CO₂ convert to CO molecules. It is important to note that the current density is not considered in the energy analysis, because current density predetermines the size and capital expense of the electrolyzers, which is outside the scope of this work.

Our baseline condition is based on Lee et al.'s report that the Ag-coated ePTFE electrode can achieve 72% CO Faradaic efficiency at -0.8 V vs. reversible hydrogen electrode in monoethanolamine aqueous solutions. We believe the current densities can be further improved if applying hydrophilic 3D porous flow-through electrodes, as very recently reported by Zhang et al.⁵² for the application of direct bicarbonate electroreduction. In the optimistic case, we anticipate the integrated electrolyser can perform similarly to the current gas-fed electrolyser. The pessimistic scenario assumes the future integrated electrolyser can only achieve a 40 % CO Faradaic efficiency at a relatively large cell potential. All these three electrolyzers are assumed to fully regenerate the capture media to a CO₂ loading at 0.3 mol CO₂/mol amine.

Table 1: Summary of CO Faradaic efficiency and cell voltages for the integrated electrolyser in different scenarios

Scenarios	CO FE (%)	Cell voltage (V)
Optimistic	90	3
Baseline	70	4
Pessimistic	40	5

In all three models, the CO₂ specific energy requirement is assumed to be 179 kJ/mol_{CO2} for amine regeneration, 16.5 kJ/mol_{CO2} for CO₂ compression, 51 kJ/mol_{CO2} PSA product separation, and 230 kJ/mol_{CO2} for bi-carbonate regeneration. All these values are based on reported literatures as listed in the Supplementary note 2.

In the sequential route, the energy consumption is shown to be dominated by CO₂ electrochemical conversion to produce CO, which includes CO₂ electrolysis (643 kJ/mol_{CO2}) and bi-carbonate regeneration (230 kJ/mol_{CO2}). The CO₂ capture requires amine regeneration energy (179 kJ/mol_{CO2}), CO₂ compression after capture (16.5 kJ/mol_{CO2}), and product purification (51 kJ/mol_{CO2}). These are all in terms of the amount of converted CO₂. Here the primary energy for the CO₂ electrolysis, compression, and product purification (based on pressure-swing adsorption) is electric work, but for bi-carbonate and amine regenerations it is inputted heat. The gas-fed CO₂ electrolyser was assumed to operate at a cell voltage of 3 V and a CO FE of 90%, which has been demonstrated experimentally (Fig. 2a). The single-pass conversion rate is assumed to be 50%, including 25% CO₂ conversion to CO and 25% CO₂ loss to bi-carbonate.

When comparing the sequential route to the baseline integrated route, there is no foreseen overall energy advantage between the two routes (Fig. 4). The primary reason is the high energy requirement to convert CO₂, which offsets any foreseen energy benefits from process intensification. Considering the higher cost for electricity than heat, the integrated route in the baseline is in fact inferior to the sequential route due to its high electrical energy consumption.

Fig. 2 Scenario analysis of overall energy cost for sequential and integrated routes. In the sequential route, the CO₂ electrolyser includes state-of-the-art gas-fed electrolysers that show 50% CO₂ utilisation or future scenarios with 100% CO₂ utilisation. The optimistic, baseline and pessimistic electrolysis cases for the integrated routes are compared against the sequential route.

In the optimistic case, we assume the electrolysis of captured CO₂ performs the same as the gas-fed electrolysis. In this scenario, the integrated route can save up to 42% of total energy due to a low cell voltage, high CO Faradaic efficiency, and no cost associated with regeneration of amines (179 kJ/molCO₂) and bi-carbonate (230 kJ/molCO₂), CO₂ compression (16.5 kJ/molCO₂), and product purification (51 kJ/molCO₂). Such reduction in energy consumption renders the integrated route a more attractive option. Our results suggest most future research emphasis to be placed on enhancing the Faradaic efficiency and cell voltages at industrially applicable current densities in order to reduce the energy of the overall process. Without these conditions, the sequential route remains favourable.

Lastly in the pessimistic case, if the integrated route has a poor CO FE (40%) and large cell voltage (5 V), however, the energy to drive integrated conversion is far higher (2412 kJ/molCO₂) than the gas-fed electrolyser, diminishing all the energy benefits from the process intensification. This scenario emphasises the importance of maximizing the two noted performance metrics.

Lastly, we assessed the energy consumption of the sequential route based on future CO₂ gas-fed electrolysis with no bi-carbonate formation. Very recent reports demonstrated the potential to improve CO₂ utilisation efficiency⁵³ by developing catalyst-membrane interface^{44,54}, optimising cell operating conditions (e.g., reducing CO₂ flow rates, increasing current densities, and optimising anolyte compositions and ionic strength)⁴⁶, or supplying protons towards the cathode to regenerate CO₂ from the bi-carbonates, e.g., flowing strong acidic catholyte^{23,55}, applying cation-exchange membranes⁴⁴ or bipolar membrane⁵⁴ in a reverse mode. The single-pass conversion rate remains 50% in this optimistic sequential model, meaning that 50% of the inputted CO₂ feed converts to CO product and reduces the required pressure-swing absorption separation energy cost. The total energy of such a sequential route is 864.5 kJ mol⁻¹ CO₂ (see Fig. 2). Here the integrated optimistic case then only maintains a maximum energy advantage of 26%. We then conclude that if the energy penalty associated with bi-carbonate regeneration is solved, there would be substantially lower energy gain possible by integrating capture and conversion even in the most optimistic scenario as described in this article.

Overall, our comparison highlights that energy benefits brought by the integrated route strongly depend on the progress in enhancing the energy efficiencies of the CO₂ electrolysis process. This trend makes sense because the CO₂ electrochemical conversion is the dominant contributor to the overall energy consumption, which is the primary reason preventing straightforward CO₂ capture and utilisation at a low cost.

On Pages 10 – 11:

The role of the single-pass conversion efficiency for the integrated electrolysis

In the analyses above, we assumed that the integrated electrolyser can fully recover the capture media to a state where it is directly recycled to the absorber (see Fig. 1a). In this section, we explore the importance of the fully recovered assumption by calculating what happens for different CO₂ loadings leaving the integrated electrolyser. In essence, this analysis examines the role of the single-pass conversion of the integrated electrolyser. Here, the CO₂ loading in the amine leaving the absorber is $X_{in} = 0.5$, while full conversion implies that the CO₂ loading leaving the electrolyser is $X_{out} = 0.3$.

The reason for the following analysis is that it may be challenging for an integrated electrolyzer to reach an outlet loading of $X_{out} = 0.3$, and we can predict what energy penalty should then be expected. For example, a sufficient concentration gradient is needed in the reactor to reach meaningful current densities (e.g., > 100 mA cm⁻²). The active species near a catalyst surface will then be depleted even if the bulk concentration remains high. Such a mass transport limitation is similar to that of H-cell CO₂ gas-fed electrolysers where CO₂ molecules transport from the liquid bulk to the electrode surface.

If the integrated electrolyser is unable recover the capture media to $X_{out} = 0.3$, there are two possible solutions. Firstly, a larger absorber size could be implemented to ensure an identical capture capacity but penalizing the processes' capital cost. In the second solution, a secondary recovery step is needed

to reach $X_{out} = 0.3$, which we analyse here. To fully recover the CO_2 -lean stream after incomplete integrated electrolysis, we have included a symbolic process (including amine regeneration, CO_2 electrolysis, product separation, and bi-carbonate regeneration, shown in Fig. 4a) to fully regenerate the capture medium and convert the rest of captured CO_2 to CO , similar to the sequential route as shown in Fig. 1a.

We find that the role of the single-pass conversion efficiency is highly dependent on the performance of the integrated electrolyser. When the electrolyser operates at the baseline conditions, the capability of the integrated electrolyser to regenerate the capture medium becomes insignificant to the energy advantage of the integrated route. In contrast, if the electrolyser operates under either optimistic or pessimistic conditions, the single-pass conversion is essential for the overall energy consumption of the integrated route. The overall energy will benefit from an efficient electrolyser with a high single-pass conversion. In contrast, a poorly performing electrolyser causes significant overall energy penalty by increasing the single-pass conversion. This observation arises from the dominant role of the electrolysis in the overall energy of the capture and conversion process.

Fig. 3 Effect of the single-pass conversion of the integrated electrolyser on the overall energy efficiency. a, A schematic illustration of the integrated route where the electrolyser is unable to fully recover the capture media. The separation and electrolysis process is symbolic process highlighted with a dashed box to regenerate the capture medium fully. X represents the CO_2 loading in the capture medium, with a unit of mol CO_2 per mol amine molecule. b, The energy comparison of the integrated route based on baseline (green solid line), pessimistic (grey), and optimistic (red) integrated electrolyser as a function of the electrolyser single-pass conversion. The grey dashed line represents the energy consumption of the sequential route based on state-of-the-art gas-fed CO_2 electrolysers. The blue region means that the integrated route is more energy-efficient than the sequential route, while the orange region indicates vice-versa.

5. Table 1 describes three scenarios and suggests a coupling between CO FE and other parameters like mol CO_2 per mol amine (the parameter X). It is presented in such a way that it implies that high FE corresponds to lowest X , and vice versa. One could have full electrochemical regeneration of amine- CO_2 (thus low X) and make entirely parasitic products, thus have very low FE, or alternatively very small currents and large X , but very high FE. This calls into question the ensuing analysis.

Response:

We agree that the scenarios as phrased may cause confusion as there is indeed no relationship between CO Faraday efficiency and the outlet loading. We have thus removed the X parameter from this part of the study to focus on the electrochemical performance metrics. We now investigate the role of X in a separate section and Fig. 4 as described by the detailed revisions to Comment 4.

6. Fig. 1 – the authors seem to have neglected pressurization needs between CO_2 capture and conversion in the sequential route.

Response:

We thank the Reviewer for this observation. We have now included the CO₂ compression in between CO₂ capture (estimated as an average of 16.5 kJ/mol CO₂ – refs 19, 20) and evaluated its role on the overall energy benefits. This factor has a minimal impact on our conclusions due to the other large energy costs. Detailed revisions can be found in below:

On Page 3:

In the amine-scrubbing cases, such a displacement could save 155 – 203 kJ/mol_{CO2} from amine regeneration^{17,19} and 14 – 19 kJ/mol_{CO2} for compression^{19,20}, which accounts for up to 90% of the total energy cost of the capture process.²¹

Fig. 1 caption: *The compression unit between stripper and electrolyzer is not shown for simplicity in the block diagram.*

On Page 8-9:

In the sequential route, the energy consumption is shown to be dominated by CO₂ electrochemical conversion to produce CO, which includes CO₂ electrolysis (643 kJ/mol_{CO2}) and bi-carbonate regeneration (230 kJ/mol_{CO2}). The CO₂ capture requires amine regeneration energy (179 kJ/mol_{CO2}), CO₂ compression after capture (16.5 kJ/mol_{CO2}), and product purification (51 kJ/mol_{CO2}). These are all in terms of the amount of converted CO₂. Here the primary energy for the CO₂ electrolysis, compression, and product purification (based on pressure-swing adsorption) is electric work, but for bi-carbonate and amine regenerations it is inputted heat. The gas-fed CO₂ electrolyser was assumed to operate at a cell voltage of 3 V and a CO FE of 90%, which has been demonstrated experimentally (Fig. 2a). The single-pass conversion rate is assumed to be 50%, including 25% CO₂ conversion to CO and 25% CO₂ loss to bi-carbonate.

When comparing the sequential route to the baseline integrated route, there is no foreseen overall energy advantage between the two routes (Fig. 4). The primary reason is the high energy requirement to convert CO₂, which offsets any foreseen energy benefits from process intensification. Considering the higher cost for electricity than heat, the integrated route in the baseline is in fact inferior to the sequential route due to its high electrical energy consumption.

Fig. 4 Scenario analysis of overall energy cost for sequential and integrated routes. In the sequential route, the CO₂ electrolyser includes state-of-the-art gas-fed electrolysers that show 50% CO₂ utilisation or future scenarios with 100% CO₂ utilisation. The optimistic, baseline and pessimistic electrolysis cases for the integrated routes are compared against the sequential route.

In the optimistic case, we assume the electrolysis of captured CO₂ performs the same as the gas-fed electrolysis. In this scenario, the integrated route can save up to 42% of total energy due to a low cell voltage, high CO Faradaic efficiency, and no cost associated with regeneration of amines (179 kJ/molCO₂) and bi-carbonate (230 kJ/molCO₂), CO₂ compression (16.5 kJ/molCO₂), and product purification (51 kJ/molCO₂).

7. The statement on p. 3, line 67 that the absence of CO₂ gas can avoid (bi)carbonate formation is not true. Many amines hydrolyze in the presence of water to yield bicarbonate – sometimes extensively more so than carbamate/ammonium, depending on the amine structure (primary/secondary vs. tertiary and sterics), pKa, time to react, etc.

Response:

Sorry for our misleading text. Here we refer to the formation of bi-carbonates in the CO₂ electrolyser unit due to the interaction of CO₂ with by-product hydroxide, a well-known issue in the electrolysis community. During high-current CO₂ electrolysis, the CO₂ can react with the hydroxide ions generated from electroreduction reaction and produce carbonate or bicarbonate. This compound cannot be used for CO₂ conversion to CO, and may also cause critical stability issues of the electrode.

We agree that the hydrolysis of carbamate can produce bicarbonate in the CO₂ capture, and heating the capture media can reverse bicarbonate back to carbamate and amines.

To clarify this point, we revised the main text on page 3 as below:

It is important to note that the formation of bicarbonate in the CO₂ absorber (usually when CO₂ loading is > 0.5 molCO₂/mol_{amine}) is not deemed as CO₂ loss, because it does not require a bi-carbonate regeneration unit to recover CO₂.

8. Overall, a major issue for this reviewer is that integrated capture and conversion is in very early stages, and as such, the tone of such an analysis, as well as delineation of how many assumptions are made and their limitations, is important. The tone of this work is more challenging and doubtful in places than is warranted given that the number of studies on this very nascent topic can be counted on one hand. I feel the work lacks scientific neutrality, by raising controversy where there is none. There is an assumption that all capture-conversion systems would, for instance, operate exactly like the one described in Lee et al., Nature Energy 2021. The authors must clearly indicate the limitations of their analysis and conclusions given how narrowly the capture-conversion technology has truly been considered in the work, and also that one or two studies should not be assumed to provide a firm foundation for performance metrics and strong debate about viability at this stage.

Response:

First, we have considered the reviewer's comments carefully in our revision, and made substantial modifications to further clarify our novelty, scientific justification and perspective as outlined in our revisions above. Where applicable, we have also modified our tone away from that of a Perspective - which acts to provide a counterbalance to recent high-profile article and reviews - to that of a neutral Article which presents previously unreported insights about the integrated route. Detailed changes to enhance scientific neutrality include:

In the abstract:

However, understanding the potential energy advantages of an integrated capture and conversion process is not straightforward due to the interconnected processes which require knowledge of both capture and electrochemical conversion processes.

On Pages 4-5:

Here we compare the operation of existing gas-fed CO₂ electrolyzers with future integrated electrolyzers. We discuss the performance metrics for both conversion processes in-depth to provide perspective on the comparative energy consumption of each route under different scenarios. We propose to gauge these two electrolyser types using the energy required to electrochemically convert one mole CO₂, which can be calculated from Eq. 1. The calculated energy is independent of the current densities, which allows us to compare these two electrolyzers despite the levels of current densities achieved in prior literature.

$$\text{Energy required to convert 1 mol CO}_2 = \frac{E \times j}{j \times FE \times \frac{1}{zF}} = \frac{E \times z \times F}{FE} \quad \text{Eq. 1}$$

where E stands for cell voltage, j for current density, FE for Faradaic efficiency, F for Faraday constant, z for the number of change to convert one CO₂ molecule (z = 2 for CO product).

On Page 5:

As a more advanced reaction, the gas-fed electrolyser outperforms the integrated electrolyser in product selectivity, current densities, and energy efficiency.^{18,32}

On Page 7:

Due to the low CO Faradaic efficiency and high cell voltage, the electrolysis of the existing early reports for captured CO₂ are at an energy cost of 10³ - 10⁵ kJ/mol_{CO2}, as compared to the 600 - 800 kJ/mol_{CO2} for the gas-fed system.

To continue the analysis in Fig. 1, we put aside the performance metrics achieved in existing integrated reports, and instead use three performance cases to see the energy comparison versus the sequential route.

Second, regarding the technological stage for this work, we feel now is a timely point to introduce greater clarity on the energy benefits of an integrated vs sequential route. For example we provide a list of 20 publications in the last few years promoting this technological direction, including 3 high-profile reviews within the past year (Sullivan *et al. Nature Catalysis*, 2021, Zhang *et al. Journal of Materials Chemistry A*, 2021, Sharifian *et al. Energy Environmental Science*, 2021) which will undoubtedly spur further research in this direction.

Importantly, our modelling results provide guidelines for the performance required of an integrated electrolyser in order for an integrated route to be less energy-intensive than a sequential route. Critically, this model is performance-based and does not utilize a specific electrolyser configuration or operating condition. The lower performance metrics displayed in Fig. 2 then provide context for reports to date but have no bearing on the modelling conclusion presented in Fig. 3.

We have made modifications throughout the manuscript:

On Page 1:

The scope of this work is limited to the CO₂ capture process based on commercially available monoethanolamine-based amine scrubbing techniques and the CO₂ electrochemical conversion to CO in gas-fed electrolysers and amine-based capture media. Shown in Fig. 1 are two comparable scenarios for a sequential capture and conversion process (Fig. 1a) and an envisioned integrated approach based on CO₂-to-CO in amine capture media (Fig. 1b).

On Page 4:

Despite a number of reports on integrated electrolysis, their current performance is inferior to the gas-fed electrolysis system owing in part to their earlier development^{18,32-36} (see Fig. 5). Regardless as the process can be evaluated as a function of performance metrics it is possible to forecast required performance targets at this early stage.

On Page 15:

Although this work is a case study over the coupled amine scrubbing and CO₂-to-CO electrochemical conversion, our simple approach is anticipated to help researchers quickly understand the upper energy limits and targeted performance metrics for different integrated CO₂ capture and electrolysis processes.

Third, our analyses are not completely based on the limited reported data of the integrated CO₂ reduction but also sourced from knowledge and insights generated from the study of CO₂ conversion in liquid for decades, such as CO₂ reduction, mass transport, product selectivity, and cell potentials. Nevertheless, we also included three more CO₂ reduction amine-based electrolytes in Fig. 1 and Table S2. Adding these data does not change our conclusions. In addition, we attempted to map out the potential energy benefits of the integrated route based on different Faradaic efficiencies and cell voltages of the integrated electrolysis, which do not have to heavily rely on the reported values.

Detailed revisions are shown below:

Table S 1 Summary of recent reports on electrochemical CO₂ reduction directly from concentrated amine solutions

Cathode	Cathode Potential (V) vs. RHE	FE_{CO} (%)	Current densities (mA cm⁻²)	Solvents	Ref
Ag/carbon-black on 300 nm Ag film on ePTFE	-0.8	72	50	30wt% MEA mixed with 2M KCl with no dissolved CO ₂	9
Ag/carbon-black on 300 nm Ag film on ePTFE	-1.2	20	100	30 % wt. MEA mixed no dissolved CO ₂	9
Ag foil	-0.8	12.4	10*	30 wt% MEA	10
Ag foil	-1.1	6.1	10*	30 wt% MEA	10
Ag foil	-1.3	2.3	10*	30 wt% MEA	10
Ag foil	-0.8	33.4	10*	30wt% MEA with 0.1% wt/wt CTAB	10
Ag foil	-1.1	15.9	10*	MEA with 0.1% wt/wt CTAB	10
Ag foil	-1.3	9.2	10*	30wt% MEA with 0.1% wt/wt CTAB	10
Cu	-0.78	45	18.4	0.1mM ethylenediamine carbamate in 0.1M NaClO₄	11
Smooth Au foil	-1.9 vs. Ag/AgCl	45	10*	2-amino-2-methyl-1-propanol (AMP) and propylene carbonate (PC) solution	12
Au/MgAl-LDHs	-0.4	68	0.8	1.0 M alcohol amine solution (n(ethanolamine): n(diethanolamine) = 2:3)	13
Cu/MgAl-LDHs	-0.25	73	0.4	1.0 M alcohol amine solution(n(ethanolamine): n(diethanolamine) = 2:3)	13
Ag	-1.1	71	15	[MEAHC][MDEA], where MEAHCl is ethanolamine hydrochloride, and MDEA is methyl diethanolamine	14

* These values are our estimations because the original paper did not report the exact values.

Fig. 5 Energy cost to convert CO_2 to CO for gas-fed CO_2 electrolyser and direct CO_2 electrochemical upgrade from capture medium. **a**, The energy cost to convert CO_2 to CO as a function of CO Faradaic efficiency with recently reported values for two different CO_2 electrolyser. Detailed data and references are summarized in Tables S2 and S3. The bubble size represents the magnitude of current densities for these cells as indicated in the legend. The insets illustrate the operating conditions of these two cells. **b**, Impacts of CO Faradaic efficiency and cell voltages on the energy cost of the CO_2 electrolyser. The solid lines indicate the Faradaic Efficiency vs. Cell voltage trends at certain energy requirements as indicated inline.

On Page 4-5:

Here we compare the operation of existing gas-fed CO_2 electrolyser with future integrated electrolyser. We discuss the performance metrics for both conversion processes in-depth to provide perspective on the comparative energy consumption of each route under different scenarios. We propose to gauge these two electrolyser types using the energy required to electrochemically convert one mole CO_2 , which can be calculated from Eq. 1. The calculated energy is independent of the current densities, which allows us to compare these two electrolyser despite the levels of current densities achieved in prior literature.

$$\text{Energy required to convert 1 mol } \text{CO}_2 = \frac{E \times j}{j \times FE \times \frac{1}{zF}} = \frac{E \times z \times F}{FE} \quad \text{Eq. 1}$$

where E stands for cell voltage, j for current density, FE for Faradaic efficiency, F for Faraday constant, z for the number of change to convert one CO_2 molecule ($z = 2$ for CO product).

9. As a general comment, the methodology, as well as various details, are generally hard to follow in the main text. For instance, Fig. 2 can only be understood by reading the text, and lacks adequate labels, references, color guides (for b), etc. Throughout the paper, energy numbers used in the model are introduced in separate paragraphs or sections, which makes it hard to follow the calculations and independently confirm the findings.

Response:

Accordingly, we have included the suggested points to improve the clarity.

On Page 2:

Fig. 6 Energy cost to convert CO_2 to CO for gas-fed CO_2 electrolyser and direct CO_2 electrochemical upgrade from capture medium. **a**, The energy cost to convert CO_2 to CO as a function of CO Faradaic efficiency with recently reported values for two different CO_2 electrolysers. Detailed data and references are summarized in Tables S2 and S3. The bubble size represents the magnitude of current densities for these cells as indicated in the legend. The insets illustrate the operating conditions of these two cells. **b**, Impacts of CO Faradaic efficiency and cell voltages on the energy cost of the CO_2 electrolysers. The solid lines indicate the Faradaic Efficiency vs. Cell voltage trends at certain energy requirements as indicated inline.

On Page 8:

Table 2: Summary of CO Faradaic efficiency and cell voltages for the integrated electrolyser in different scenarios

Scenarios	CO FE (%)	Cell voltage (V)
Optimistic	90	3
Baseline	70	4
Pessimistic	40	5

In all three models, the CO_2 specific energy requirement is assumed to be 179 $\text{kJ/mol}_{\text{CO}_2}$ for amine regeneration, 16.5 $\text{kJ/mol}_{\text{CO}_2}$ for CO_2 compression, 51 $\text{kJ/mol}_{\text{CO}_2}$ PSA product separation, and 230 $\text{kJ/mol}_{\text{CO}_2}$ for bi-carbonate regeneration. All these values are based on reported literatures as listed in the Supplementary note 2.

In the sequential route, the energy consumption is shown to be dominated by CO_2 electrochemical conversion to produce CO , which includes CO_2 electrolysis (643 $\text{kJ/mol}_{\text{CO}_2}$) and bi-carbonate regeneration (230 $\text{kJ/mol}_{\text{CO}_2}$). The CO_2 capture requires amine regeneration energy (179 $\text{kJ/mol}_{\text{CO}_2}$), CO_2 compression after capture (16.5 $\text{kJ/mol}_{\text{CO}_2}$), and product purification (51 $\text{kJ/mol}_{\text{CO}_2}$). These are all in terms of the amount of converted CO_2 . Here the primary energy for the CO_2 electrolysis, compression, and product purification (based on pressure-swing adsorption) is electric work, but for bi-carbonate and amine regenerations it is inputted heat.

On Page 9:

In the optimistic case, we assume the electrolysis of captured CO_2 performs the same as the gas-fed electrolysis. In this scenario, the integrated route can save up to 42% of total energy due to a low cell voltage, high CO Faradaic efficiency, and no cost associated with regeneration of amines (179 $\text{kJ/mol}_{\text{CO}_2}$) and bi-carbonate (230 $\text{kJ/mol}_{\text{CO}_2}$), CO_2 compression (16.5 $\text{kJ/mol}_{\text{CO}_2}$), and product purification (51 $\text{kJ/mol}_{\text{CO}_2}$).

Reviewer #2

This work present integrated CO₂ capture and electrochemical CO₂ conversion processes and compare it with conventional sequential process. The topic of the study is appropriate for the journal because the amine captured CO₂ reduction has great potential that eliminate energy intensive CO₂ stripping processes and product separation process. However I do not support the publication of this manuscript in Nature Communications.

First of all this study only focuses on the energy analysis rather than comprehensive techoneconomic analysis. As numbers of studies indicate, electrochemical process include expensive unit operations such as electrolyzer, thus the capital investment cost cannot be overlooked. Authors emphasize the high FE throughout the paper but current density (related to size of electrolyzer) can be also critical issue for the practical application. I also do not agree with the carbonate stripper in both sequential and integrated root in figure

Response:

First of all, we would like to appreciate Reviewer #2's efforts in reviewing our work.

As the Reviewer noticed, our work is not a complete techno-economic paper but rather a perspective from the energy point of view. As mentioned in Reviewer #1's Comment 8 and our main text, the study over the integrated electrolyser is at the early stage, which lacks sufficient data for the cell structures, current densities and cell potentials. We are afraid including the capital cost analyses will only bring further complexity and uncertainty to our analyses. Here we are communicating more directly to the electrolysis community up taking such research efforts, which uses energy as a motivator independent of techno-economics. We agree that both are essential, but here techno-economics are outside of the scope of our work.

We have simplified our analysis to initially assume that the integrated route does not require further CO₂ stripping prior to returning to the absorber. Detailed discussion on this is provided in the response to Reviewer 1 Comment 4.

1. Normally lean amine loading for the MEA CO₂ capture system ranges between 0.2-0.3 thus futher CO₂ stripping may exaggerate the energy requirement. Also process detail of the systems are not presented in this work, thus I cannot evaluate the credibility of the model. Although author did comprehensive analysis for the energy consumption but I think different types of energy cannot not be compared the same criteria. (e.g., thermal and electricity should compare in terms of cost not amount). Most importantly, carbamate reduction for the electrolysis of captured CO₂ electrolysis is very questionable. Numbers of studies indicate strong C-N bond in the carbamate hinders direct conversion of CO₂, and addition of potassium ions or caesium ions may result additional capital and operational cost due to their recovery and recycle. However, authors did not discuss the process supplying cations and thier recycling.

Response:

CO₂ loading and different energy comparison

The reviewer has made a good point. We have corrected the CO₂ loadings of the amine solutions in the models and discussed the heat and electrical energy when comparing different scenarios. Detailed revisions are as below:

On page 3:

Fig. 7 Sequential and integrated routes of CO₂ capture and conversion. **a**, Schematic illustration and block diagrams of the sequential route for amine-based CO₂ capture and electrolysis to produce CO. CO₂ electrolyser is based on membrane-electrode assemblies. The compression unit between stripper and electrolyzer is not shown for simplicity in the block diagram. **b**, Schematic illustration and block flow diagrams of integrated CO₂ capture and direct CO₂ electroreduction from capture medium. The CO₂ loading of the CO₂-rich and CO₂-lean amine streams are assumed based on Gjernes et al. report¹⁷.

On Pages 7-10

With the conversion processes described for the sequential and integrated routes, we can compare the expected energy requirement for both routes shown in Fig. 1 through a mass and energy balance. Detailed description of the models can be found in Supplementary Note 2.

Here Fig. 2 explores the potential energy advantages of the integrated route under optimistic, baseline, and pessimistic performance metric scenarios for the electrolysis processes. Detailed conditions for these scenarios are summarized in Table 1 using the two most critical parameters for the integrated electrolysis process: CO Faradaic efficiency and cell voltage. The sequential route cases assume the gas-fed electrolyser to be operated at 3 V, 90% CO Faraday efficiency, and 50% single pass conversion. The with-bi-carbonate case assumes 50% of the reacted CO₂ convert to bicarbonate, while the without-bi-carbonate case assumes all the reacted CO₂ convert to CO molecules. It is important to note that the current density is not considered in the energy analysis, because current density predetermines the size and capital expense of the electrolyzers, which is outside the scope of this work.

Our baseline condition is based on Lee et al.'s report that the Ag-coated ePTFE electrode can achieve 72% CO Faradaic efficiency at -0.8 V vs. reversible hydrogen electrode in monoethanolamine aqueous solutions. We believe the current densities can be further improved if applying hydrophilic 3D porous flow-through electrodes, as very recently reported by Zhang et al.⁵² for the application of direct bicarbonate electroreduction. In the optimistic case, we anticipate the integrated electrolyser can perform similarly to the current gas-fed electrolyser. The pessimistic scenario assumes the future integrated electrolyser can only achieve a 40 % CO Faradaic efficiency at a relatively large cell potential. All these three electrolyzers are assumed to fully regenerate the capture media to a CO₂ loading at 0.3 mol CO₂/ mol amine.

Table 3: Summary of CO Faradaic efficiency and cell voltages for the integrated electrolyser in different scenarios

Scenarios	CO FE (%)	Cell voltage (V)
Optimistic	90	3
Baseline	70	4
Pessimistic	40	5

In all three models, the CO₂ specific energy requirement is assumed to be 179 kJ/mol_{CO2} for amine regeneration, 16.5 kJ/mol_{CO2} for CO₂ compression, 51 kJ/mol_{CO2} PSA product separation, and 230 kJ/mol_{CO2} for bi-carbonate regeneration. All these values are based on reported literatures as listed in the Supplementary note 2.

In the sequential route, the energy consumption is shown to be dominated by CO₂ electrochemical conversion to produce CO, which includes CO₂ electrolysis (643 kJ/mol_{CO2}) and bi-carbonate regeneration (230 kJ/mol_{CO2}). The CO₂ capture requires amine regeneration energy (179 kJ/mol_{CO2}), CO₂ compression after capture (16.5 kJ/mol_{CO2}), and product purification (51 kJ/mol_{CO2}). These are all in terms of the amount of converted CO₂. Here the primary energy for the CO₂ electrolysis, compression, and product purification (based on pressure-swing adsorption) is electric work, but for bi-carbonate and amine regenerations it is inputted heat. The gas-fed CO₂ electrolyser was assumed to operate at a cell voltage of 3 V and a CO FE of 90%, which has been demonstrated experimentally (Fig. 2a). The single-pass conversion rate is assumed to be 50%, including 25% CO₂ conversion to CO and 25% CO₂ loss to bi-carbonate.

When comparing the sequential route to the baseline integrated route, there is no foreseen overall energy advantage between the two routes (Fig. 4). The primary reason is the high energy requirement to convert CO₂, which offsets any foreseen energy benefits from process intensification. Considering the higher cost for electricity than heat, the integrated route in the baseline is in fact inferior to the sequential route due to its high electrical energy consumption.

Fig. 8 Scenario analysis of overall energy cost for sequential and integrated routes. In the sequential route, the CO₂ electrolyser includes state-of-the-art gas-fed electrolysers that show 50% CO₂ utilisation or future scenarios with 100% CO₂ utilisation. The optimistic, baseline and pessimistic electrolysis cases for the integrated routes are compared against the sequential route.

In the optimistic case, we assume the electrolysis of captured CO₂ performs the same as the gas-fed electrolysis. In this scenario, the integrated route can save up to 42% of total energy due to a low cell voltage, high CO Faradaic efficiency, and no cost associated with regeneration of amines (179 kJ/molCO₂) and bi-carbonate (230 kJ/molCO₂), CO₂ compression (16.5 kJ/molCO₂), and product purification (51 kJ/molCO₂). Such reduction in energy consumption renders the integrated route a more attractive option. Our results suggest most future research emphasis to be placed on enhancing the Faradaic efficiency and cell voltages at industrially applicable current densities in order to reduce the energy of the overall process. Without these conditions, the sequential route remains favourable.

Lastly in the pessimistic case, if the integrated route has a poor CO FE (40%) and large cell voltage (5 V), however, the energy to drive integrated conversion is far higher (2412 kJ/molCO₂) than the gas-fed electrolyser, diminishing all the energy benefits from the process intensification. This scenario emphasises the importance of maximizing the two noted performance metrics.

Lastly, we assessed the energy consumption of the sequential route based on future CO₂ gas-fed electrolysis with no bi-carbonate formation. Very recent reports demonstrated the potential to improve CO₂ utilisation efficiency⁵³ by developing catalyst-membrane interface^{44,54}, optimising cell operating conditions (e.g., reducing CO₂ flow rates, increasing current densities, and optimising anolyte compositions and ionic strength)⁴⁶, or supplying protons towards the cathode to regenerate CO₂ from the bi-carbonates, e.g., flowing strong acidic catholyte^{23,55}, applying cation-exchange membranes⁴⁴ or bipolar membrane⁵⁴ in a reverse mode. The single-pass conversion rate remains 50% in this optimistic sequential model, meaning that 50% of the inputted CO₂ feed converts to CO product and reduces the required pressure-swing absorption separation energy cost. The total energy of such a sequential route is 864.5 kJ mol⁻¹ CO₂ (see Fig. 2). Here the integrated optimistic case then only maintains a maximum energy advantage of 26%. We then conclude that if the energy penalty associated with bi-carbonate regeneration is solved, there would be substantially lower energy gain possible by integrating capture and conversion even in the most optimistic scenario as described in this article.

Overall, our comparison highlights that energy benefits brought by the integrated route strongly depend on the progress in enhancing the energy efficiencies of the CO₂ electrolysis process. This trend makes sense because the CO₂ electrochemical conversion is the dominant contributor to the overall energy consumption, which is the primary reason preventing straightforward CO₂ capture and utilisation at a low cost.

On Pages 10 – 11:

The role of the single-pass conversion efficiency for the integrated electrolysis

In the analyses above, we assumed that the integrated electrolyser can fully recover the capture media to a state where it is directly recycled to the absorber (see Fig. 1a). In this section, we explore the importance of the fully recovered assumption by calculating what happens for different CO₂ loadings leaving the integrated electrolyser. In essence, this analysis examines the role of the single-pass conversion of the integrated electrolyser. Here, the CO₂ loading in the amine leaving the absorber is $X_{in} = 0.5$, while full conversion implies that the CO₂ loading leaving the electrolyser is $X_{out} = 0.3$.

The reason for the following analysis is that it may be challenging for an integrated electrolyzer to reach an outlet loading of $X_{out} = 0.3$, and we can predict what energy penalty should then be expected. For example, a sufficient concentration gradient is needed in the reactor to reach meaningful current densities (e.g., > 100 mA cm⁻²). The active species near a catalyst surface will then be depleted even if the bulk concentration remains high. Such a mass transport limitation is similar to that of H-cell CO₂ gas-fed electrolysers where CO₂ molecules transport from the liquid bulk to the electrode surface.

If the integrated electrolyser is unable recover the capture media to $X_{out} = 0.3$, there are two possible solutions. Firstly, a larger absorber size could be implemented to ensure an identical capture capacity but penalizing the processes' capital cost. In the second solution, a secondary recovery step is needed

to reach $X_{out} = 0.3$, which we analyse here. To fully recover the CO_2 -lean stream after incomplete integrated electrolysis, we have included a symbolic process (including amine regeneration, CO_2 electrolysis, product separation, and bi-carbonate regeneration, shown in Fig. 4a) to fully regenerate the capture medium and convert the rest of captured CO_2 to CO , similar to the sequential route as shown in Fig. 1a.

We find that the role of the single-pass conversion efficiency is highly dependent on the performance of the integrated electrolyser. When the electrolyser operates at the baseline conditions, the capability of the integrated electrolyser to regenerate the capture medium becomes insignificant to the energy advantage of the integrated route. In contrast, if the electrolyser operates under either optimistic or pessimistic conditions, the single-pass conversion is essential for the overall energy consumption of the integrated route. The overall energy will benefit from an efficient electrolyser with a high single-pass conversion. In contrast, a poorly performing electrolyser causes significant overall energy penalty by increasing the single-pass conversion. This observation arises from the dominant role of the electrolysis in the overall energy of the capture and conversion process.

Fig. 9 Effect of the single-pass conversion of the integrated electrolyser on the overall energy efficiency. a, A schematic illustration of the integrated route where the electrolyser is unable to fully recover the capture media. The separation and electrolysis process is symbolic process highlighted with a dashed box to regenerate the capture medium fully. X represents the CO_2 loading in the capture medium, with a unit of mol CO_2 per mol amine molecule. b, The energy comparison of the integrated route based on baseline (green solid line), pessimistic (grey), and optimistic (red) integrated electrolyser as a function of the electrolyser single-pass conversion. The grey dashed line represents the energy consumption of the sequential route based on state-of-the-art gas-fed CO_2 electrolysers. The blue region means that the integrated route is more energy-efficient than the sequential route, while the orange region indicates vice-versa.

Carbamate reduction for CO_2 conversion

There are still debates on the primary catalytically active species for CO_2 electroreduction. To improve the accuracy, we have changed the terms related to direct carbamate reduction to CO_2 conversion and expanded our discussions in the outlook about the need to understand the primary active species for CO_2 reduction. Detailed revisions are as below:

On Pages 12-13:

What are the primary catalytically active species?

It has been reported recently that the catalysts for gas-fed CO_2 electroreduction are selective to reduce CO_2 captured by amine-based capture media (RNH_2).^{18,32,33,57} In the CO_2 -rich amines, the zwitterions ions including $RNHCO_2^-$ and RNH_3^+ are the major CO_2 species in the case of 30 wt%

monoethanolamine aqueous solution when the CO_2 loading is below 0.4 – 0.6 mol CO_2 per mol amine.^{58,59} Further increase of CO_2 loadings could promote carbamate hydrolysis to produce bi-carbonates. Therefore, the CO_2 associated species should include carbamate ions, bi-carbonate ions, and minor free dissolved CO_2 , all may contribute to the CO_2 conversion.

However, there are still debates on the primary catalytically active species for the conversion in the amine (particularly for monoethanolamine) solutions. (see Fig. 5a) Early report by Chen et al.³² claims that the free CO_2 dissolved in water can be the primary active species for the conversion, with nearly 100% Faradaic efficiency of hydrogen evolution regardless of the carbamate concentrations. In contrast, recent reports argued the possibility to reduce the carbamate ions as the main active reagent.^{18,60} The claimed mechanisms for the direct carbamate reduction is different from the reduction mechanisms in CO_2 electrolysis⁴⁰ and direct bicarbonate reduction^{61,62}. Interestingly, these recent reports also show an improvement of CO_2 conversion selectivity by increasing operating temperatures¹⁸ or including large alkali cations (e.g., Cs^+)⁶⁰, both actually help destabilise the carbamate and thus release free CO_2 . Therefore, the primary catalytically reactant for CO_2 conversion still remains a mystery, but is paramount for the rational development of an efficient electrochemical system for integration.

In the CO_2 capture step based on 30 wt% monoethanolamine solutions, the CO_2 loadings are usually at 0.3 – 0.5 mole CO_2 per mole amine, meaning that the concentration of the bi-carbonate and free CO_2 are negligible. If the free CO_2 is the primary active reagent, regenerating and concentrating free CO_2 from carbamate and bicarbonate should be the key step to improve the integrated CO_2 conversion. Meanwhile, this strategy could adversely impact CO_2 capture. If the carbamate ions are the primary catalytically active species, they could be repelled by the negatively charged cathode surface, which might limit the coverage of reactants, especially at high overpotentials. For example, Khurram et al.⁶⁰ reported that including alkali cations such as K^+ and Cs^+ could promote CO_2 conversion but may destabilize the formation of carbamate, which is essential for CO_2 absorption. The results of our energy analysis indicate that the capture media for the integrated route could be designed to favour CO_2 conversion at a reasonable cost on CO_2 absorption. Therefore, an interdisciplinary collaboration between CO_2 capture and electrolysis is highly important to advance the integrated route.

Fig. 10 Speciation of amine-based capture media in sequential and integrated routes and their impacts on CO_2 electrochemical conversion. a, Proposed integrated CO_2 absorption and electrolysis routes in amine-based solvents. b, Schematic illustration of the role of alkali cations which promote interfacial charge transfer from the catalyst surface to the carbamate ions, adapted from¹⁸.

Cation recycling

We believe cations should be present in the capture media through CO_2 capture and conversion, and there is no need for cation recycling. That is why we discussed the potential penalty on the CO_2 absorption on Page 13. See below:

If the carbamate ions are the primary catalytically active species, they could be repelled by the negatively charged cathode surface, which might limit the coverage of reactants, especially at high overpotentials. For example, Khurram et al.⁶⁰ reported that including alkali cations such as K^+ and Cs^+ could promote CO_2 conversion but may destabilize the formation of carbamate, which is essential for CO_2 absorption. The results of our energy analysis indicate that the capture media for the integrated

route could be designed to favour CO₂ conversion at a reasonable cost on CO₂ absorption. Therefore, an interdisciplinary collaboration between CO₂ capture and electrolysis is highly important to advance the integrated route.

Detail comments are listed below. English Correction: first paragraph on page 2.

Below are the parts found in paragraph

1. - CO₂ capture & conversion “is”... - CO₂ capture can (be) operate(d) processes are also now operating (now being operated??) - Low-temperature CO₂ electrolysis using pure CO₂ feeds “have” (not has)...

Response:

These typos have been corrected. See below:

On Page 2:

Carbon dioxide (CO₂) capture and subsequent conversion represents a promising route for the production of fossil-fuel-free fuels and feedstocks from waste CO₂.

For example, CO₂ capture can be operated at an overall cost of US\$50 – 150 to capture one tonne CO₂

On the conversion side, low-temperature CO₂ electrolyzers using pure CO₂ feeds have achieved a current beyond 1 A cm⁻² to convert CO₂ selectively to feedstocks (e.g., carbon monoxide (CO) and ethylene (C₂H₄)).⁷⁻¹⁰

2. In page 5 “In molecular CO₂ reduction, dissolved CO₂ is the main catalytically reactant for the conversion” needs to rephrase.

Response:

Now the corrected sentence is on Page 6:

In gas-fed CO₂ electroreduction, the dissolved CO₂ in water is the main catalytically reactant for the conversion.^{40,41}

3. Page 7 Table 1: I wonder why the value for current density is not mentioned. Also, I'm curious how to choose scenario for X. both pessimistic and optimistic cases show lower X than baseline and it is very confusing. Corresponding result in Fig 3 b the base case energy penalty for regeneration is the highest in the base case. Please address proper logic behind of it In addition, in the middle paragraph of page 6, referring to Ref 17 (Nature Energy in 2021) as an example of the electrochemical reactor, it was 3.7V, 100 mA/cm², FE 20% (CO). but the baseline FE is set at 70%. I'd like to know the rationale too.

Response:

Comments on current density

Sorry for the confusion. We used an energy consumption term that is not dependent on current densities. However, we acknowledge that current density is an important factor that predetermines size of the electrolyser, the eventual cell voltage and capital expenses. Here in our primary analysis we have chosen not to make assumptions regarding current density limitations, leaving room for unforeseen innovations that increase current density in an integrated electrolyser. For clarification, we added further discussion:

On Pages 4-5:

We propose to gauge these two electrolyser types using the energy required to electrochemically convert one mole CO₂, which can be calculated from Eq. 1. The calculated energy is independent of the current densities, which allows us to compare these two electrolysers despite the levels of current densities achieved in prior literature.

$$\text{Energy required to convert 1 mol CO}_2 = \frac{E \times j}{j \times FE \times \frac{1}{zF}} = \frac{E \times z \times F}{FE} \quad \text{Eq. 1}$$

where E stands for cell voltage, j for current density, FE for Faradaic efficiency, F for Faraday constant, z for the number of change to convert one CO₂ molecule (z = 2 for CO product).

On Pages 6-7:

The catalytically active species for CO₂ conversion in the amine solutions remain unclear but should be related to the free CO₂ dissolved in the solution, carbamate and bicarbonate ions, partly or all present in a CO₂-rich amine stream. These CO₂ species then need to diffuse to the negatively electrode through a boundary layer usually > 40 μm, similar to a CO₂-fed aqueous H-cell electrolysers.^{48,49}

On Page 7:

It is important to note that the current density is not considered in the energy analysis, because current density predetermines the size and capital expense of the electrolysers, which is outside the scope of this work.

On Page 12:

Like the gas-fed CO₂ electrolysers, we believe operating at more than 100 mA cm⁻² with a high product selectivity is a prerequisite for an industrially relevant integrated system.⁵⁶

X in the scenario analyses

Sorry again for the confusion. For clarification, our initial analysis described in Fig. 1 now assumes the electrolyser can achieve CO₂-lean amines with no extra processes for regeneration and conversion. We then provide the implications of X and single-pass conversion efficiency in a separate section and Fig. 4.

The detailed revisions are:

On Pages 7-10:

With the conversion processes described for the sequential and integrated routes, we can compare the expected energy requirement for both routes shown in Fig. 1 through a mass and energy balance. Detailed description of the models can be found in Supplementary Note 2.

Here Fig. 2 explores the potential energy advantages of the integrated route under optimistic, baseline, and pessimistic performance metric scenarios for the electrolysis processes. Detailed conditions for these scenarios are summarized in Table 1 using the two most critical parameters for the integrated electrolysis process: CO Faradaic efficiency and cell voltage. The sequential route cases assume the gas-fed electrolyser to be operated at 3 V, 90% CO Faraday efficiency, and 50% single pass conversion. The with-bi-carbonate case assumes 50% of the reacted CO₂ convert to bicarbonate, while the without-bi-carbonate case assumes all the reacted CO₂ convert to CO molecules. It is important to note that the current density is not considered in the energy analysis, because current density predetermines the size and capital expense of the electrolysers, which is outside the scope of this work.

Our baseline condition is based on Lee et al.'s report that the Ag-coated ePTFE electrode can achieve 72% CO Faradaic efficiency at -0.8 V vs. reversible hydrogen electrode in monoethanolamine aqueous solutions. We believe the current densities can be further improved if applying hydrophilic 3D porous flow-through electrodes, as very recently reported by Zhang et al.⁵² for the application of direct bicarbonate electroreduction. In the optimistic case, we anticipate the integrated electrolyser can perform similarly to the current gas-fed electrolyser. The pessimistic scenario assumes the future integrated electrolyser can only achieve a 40 % CO Faradaic efficiency at a relatively large cell potential. All these three electrolysers are assumed to fully regenerate the capture media to a CO₂ loading at 0.3 mol CO₂/mol amine.

Table 4: Summary of CO Faradaic efficiency and cell voltages for the integrated electrolyser in different scenarios

Scenarios	CO FE (%)	Cell voltage (V)
Optimistic	90	3
Baseline	70	4
Pessimistic	40	5

In all three models, the CO₂ specific energy requirement is assumed to be 179 kJ/mol_{CO₂} for amine regeneration, 16.5 kJ/mol_{CO₂} for CO₂ compression, 51 kJ/mol_{CO₂} PSA product separation, and 230 kJ/mol_{CO₂} for bi-carbonate regeneration. All these values are based on reported literatures as listed in the Supplementary note 2.

In the sequential route, the energy consumption is shown to be dominated by CO₂ electrochemical conversion to produce CO, which includes CO₂ electrolysis (643 kJ/mol_{CO₂}) and bi-carbonate regeneration (230 kJ/mol_{CO₂}). The CO₂ capture requires amine regeneration energy (179 kJ/mol_{CO₂}), CO₂ compression after capture (16.5 kJ/mol_{CO₂}), and product purification (51 kJ/mol_{CO₂}). These are all in terms of the amount of converted CO₂. Here the primary energy for the CO₂ electrolysis, compression, and product purification (based on pressure-swing adsorption) is electric work, but for bi-carbonate and amine regenerations it is inputted heat. The gas-fed CO₂ electrolyser was assumed to operate at a cell voltage of 3 V and a CO FE of 90%, which has been demonstrated experimentally (Fig. 2a). The single-pass conversion rate is assumed to be 50%, including 25% CO₂ conversion to CO and 25% CO₂ loss to bi-carbonate.

When comparing the sequential route to the baseline integrated route, there is no foreseen overall energy advantage between the two routes (Fig. 4). The primary reason is the high energy requirement to convert CO₂, which offsets any foreseen energy benefits from process intensification. Considering the higher cost for electricity than heat, the integrated route in the baseline is in fact inferior to the sequential route due to its high electrical energy consumption.

Fig. 11 Scenario analysis of overall energy cost for sequential and integrated routes. In the sequential route, the CO₂ electrolyser includes state-of-the-art gas-fed electrolysers that show 50% CO₂ utilisation or future scenarios with 100% CO₂ utilisation. The optimistic, baseline and pessimistic electrolysis cases for the integrated routes are compared against the sequential route.

In the optimistic case, we assume the electrolysis of captured CO₂ performs the same as the gas-fed electrolysis. In this scenario, the integrated route can save up to 42% of total energy due to a low cell voltage, high CO Faradaic efficiency, and no cost associated with regeneration of amines (179 kJ/molCO₂) and bi-carbonate (230 kJ/molCO₂), CO₂ compression (16.5 kJ/molCO₂), and product purification (51 kJ/molCO₂). Such reduction in energy consumption renders the integrated route a more attractive option. Our results suggest most future research emphasis to be placed on enhancing the Faradaic efficiency and cell voltages at industrially applicable current densities in order to reduce the energy of the overall process. Without these conditions, the sequential route remains favourable.

Lastly in the pessimistic case, if the integrated route has a poor CO FE (40%) and large cell voltage (5 V), however, the energy to drive integrated conversion is far higher (2412 kJ/molCO₂) than the gas-fed electrolyser, diminishing all the energy benefits from the process intensification. This scenario emphasises the importance of maximizing the two noted performance metrics.

The rationale for the baseline condition

We in fact used the CO Faraday efficiency at 50 mA cm⁻² as the baseline, which we expect can be further improved significantly by applying hydrophilic flow-through electrodes. To improve the clarity, we added the following texts on Page 7.

Our baseline condition is based on Lee et al.'s report that the Ag-coated ePTFE electrode can achieve 72% CO Faradaic efficiency at -0.8 V vs. reversible hydrogen electrode in monoethanolamine aqueous solutions. We believe the current densities can be further improved if applying hydrophilic 3D porous flow-through electrodes, as very recently reported by Zhang et al.⁵² for the application of direct bicarbonate electroreduction. In the optimistic case, we anticipate the integrated electrolyser can perform similarly to the current gas-fed electrolyser. The pessimistic scenario assumes the future integrated electrolyser can only achieve a 40% CO Faradaic efficiency at a relatively large cell potential. All these three electrolysers are assumed to fully regenerate the capture media to a CO₂ loading at 0.3 mol CO₂/mol amine.

4. Eq.2-4 on page 8 of SI: The formula does not make sense. $F_{in}X$ is the amount of (X) converted in $CO_2(F_{in})$ entering the electrochemical reactor, which should be $F_{in} - F_p$, where $F_{in}-F_s=2F_p$ in the formula. Perhaps the definition of conversion means electrochemical reduction & permeation through MEA. It would be good to express clearly in the body and SI.

Response:

As we mentioned in Table S4, the single pass conversion indicates the conversion of CO_2 to both CO and carbonate. F_s indicates the unreacted CO_2 in the effluent. According to CO_2 balance, $F_{in}(1-x) = F_s$. We further clarified the CO_2 balance in the supplementary note 2 on Page 9:

We assumed the single pass conversion x of CO_2 in the electrolyser is 0.5, so F_{in} and F_s indicate the CO_2 inlet and outlet flow rates, respectively. Therefore, the CO_2 balance is:

$$F_{in} - F_s = F_{in}x \quad \text{Eq. 2-4}$$

5. SI page 9 Eq. 2-10 Equation following: Equation number is missing. Also, authors said $Q_c=Q_{sc}$ because $F_c=F_o$, but since both Q_c and Q_{sc} have units of kJ/mol, it does not seem to be related to F , the flow rate.

Response:

We have now updated these equation numbers.

These energy terms are the energies required to capture and convert one mole CO_2 converted. We now show the full equations for all the energy calculations in the supplementary information for clarification:

One example is shown in below on Page 10.

Because of $F_c = F_o$ in our assumption, the energy to recover one molar CO_2 from the (bi)carbonate is:

$$Q_c = Q_{sc} \times \frac{F_c}{F_o} \quad \text{Eq. 2-11}$$

6. SI page 9 Eq.2-12: It said that PSA is a major energy item, but Q_{psa} (Eq. 2-10) of the PSA process is not included.

Response:

We in fact included this in the model but forgot to include it in the Supplementary information. Now we have now included it on Page 9.

$$Q_{psa} = Q_{sspa} \times \frac{F_s}{F_o} \quad \text{Eq. 2-10}$$

7 SI page 11 I recommend recheck carbon balance equation for example Eq. 2-16 seems $F_{ps}=F_a*X-F_a*X_{out} = F_a(X-X_{out})$. Why author divide $(X_{in}-X_{out})$? To be sure, it is strongly recommended that authors disclose code/excel files etc.

Response:

It is the same issue to Comment 5. $F_a(X-X_{out})$ is the CO_2 molar flow rate that is converted in the integrated electrolyser, while $F_a(X_{in}-X_{out})$ is the total CO_2 captured and needs to be converted. So the

energy equation divided by $F_a(X_{in}-X_{out})$ can obtain the energy required to capture & convert one mole CO_2 . We have now corrected them, see below.

On Page 12 in the supplementary information:

The energy required to convert one mole CO_2 in the amine solutions in the integrated electrolysis:

$$Q_{e_integrated} = \frac{F_a}{F_a} \times \frac{X_{in} - X}{X_{in} - X_{out}} \times \frac{E_{integrated} \times j}{j \times FE_{integrated}/(z \times F)} \quad Eq. 2-18$$

The energy required to capture and convert the CO_2 in the normal electrolyser in the separation & electrolysis process:

$$Q_{e_integrated_s} = \frac{F_a}{F_a} \times \frac{X - X_{out}}{X_{in} - X_{out}} \times \frac{E \times j}{j \times FE/(z \times F)} \quad Eq. 2-19$$

The energy required for amine regeneration in the overall process is:

$$Q_{ar_integrated} = \frac{F_a}{F_a} \times Q_{sar} + Q_{ar} \times \frac{F_a}{F_a} \times \frac{X - X_{out}}{X_{in} - X_{out}} \quad Eq. 2-20$$

The energy required for bi-carbonate regeneration process is:

$$Q_{c_integrated} = \frac{F_a}{F_a} \times \frac{X - X_{out}}{X_{in} - X_{out}} \times Q_c \quad Eq. 2-21$$

Appendix: Chronological list of recent publications in the CO₂ electroreduction in capture media

No.	Paper Title
1	Chen et al. Electrochemical Reduction of Carbon Dioxide in a Monoethanolamine Capture Medium, ChemSusChem , 2017, 10, 4109.
2	Filotás et al., Extended Investigation of Electrochemical CO ₂ Reduction in Ethanolamine Solutions by SECM, Electroanalysis , 2018, 30, 690.
3	Diaz et al. Electrochemical production of syngas from CO ₂ captured in switchable polarity solvents, Green Chemistry , 2018, 20, 620-626.
4	Machado and Ponte, CO ₂ capture and electrochemical conversion, Current Opinion in Green and Sustainable Chemistry , 2018, 86-90.
5	Khurram et al., Tailoring the Discharge Reaction in Li-CO ₂ Batteries through Incorporation of CO ₂ Capture Chemistry, Joule , 2018, 2, 2649-2666.
6	Garg et al., Catalyst–Electrolyte Interactions in Aqueous Reline Solutions for Highly Selective Electrochemical CO ₂ Reduction, ChemSusChem , 2019, 13, 304-311.
7	Khurram et al, Promoting Amine-Activated Electrochemical CO ₂ Conversion with Alkali Salts, Journal of Physical Chemistry C , 2019, 18222-18231.
8	Abdinejad et al, Enhanced Electrocatalytic Activity of Primary Amines for CO ₂ Reduction Using Copper Electrodes in Aqueous Solution, ACS Sustainable Chemistry & Engineering , 2020, 8, 1715-1720.
9	Bhattacharya et al, Toward Combined Carbon Capture and Recycling: Addition of an Amine Alters Product Selectivity from CO to Formic Acid in Manganese Catalyzed Reduction of CO ₂ , Journal of the American Chemical Society , 2020, 142, 17589-17597.
10	Hossain et al, Electrochemical Reduction of CO ₂ at Coinage Metal Nanodendrites in Aqueous Ethanolamine, Chemistry A European Journal , 2020, 27, 1346-1355.
11	Khurram et al, Effects of Temperature on Amine-Mediated CO ₂ Capture and Conversion in Li Cells, Journal of Physical Chemistry C , 2020, 124, 18877-18885.
12	Lee et al, Electrochemical upgrade of CO ₂ from amine capture solution, Nature Energy , 6, 46-53.
13	Goetheer et al, Integrating CO ₂ Capture with Electrochemical Conversion Using Amine-Based Capture Solvents as Electrolytes, Industrial & Engineering Chemistry Research , 2021, 60, 4269-4278.
14	Ahmad et al, Electrochemical CO ₂ reduction to CO facilitated by MDEA-based deep eutectic solvent in aqueous solution, Renewable Energy , 2021, 177, 23-33.
15	Sullivan et al., Coupling electrochemical CO ₂ conversion with CO ₂ capture, Nature Catalysis , 4, 952-958.
16	Zhang et al., Materials and system design for direct electrochemical CO ₂ conversion in capture media, Journal of Materials Chemistry A , 2021,9, 18785-18792.
17	Sharifian et al., Electrochemical carbon dioxide capture to close the carbon cycle, Energy & Environmental Science , 2021,14, 781-814.
18	Li et al., Electrolytic reduction of CO ₂ in KHCO ₃ and alkanolamine solutions with layered double hydroxides intercalated with gold or copper, Electrochimica Acta , 2022, 402, 139523.
19	Zhang et al., Porous metal electrodes enable efficient electrolysis of carbon capture solutions, Energy & Environmental Science , 2022, ,15, 705-713.
20	Gutiérrez-Sánchez et al., A State-of-the-Art Update on Integrated CO ₂ Capture and Electrochemical Conversion Systems, ChemElectroChem , 2022, 9, e202101540.

REVIEWER COMMENTS

Reviewer #1 (Remarks to the Author):

This version of the manuscript is significantly improved and provides more nuanced discussion. The authors have taken care to carefully justify their reasoning and numbers used in many places, making it easier to evaluate the assumptions and findings. The current manuscript is more rigorous and could be considered for publication pending the ability to adequately address remaining yet still significant key issues below. Some of these are points that the reviewer continues to challenge and would like to see elaborated and better justified.

1. (Relatively minor comment) In the revised manuscript the authors have updated their definition of full regeneration to now refer to 0.3 loading. While this is commended and, the reviewer feels, quite appropriate operationally, the language may still be confusing to readers who may expect "full" to refer to 0 loading. The authors are recommended to polish this term, maybe to just use "lean loading state" consistently rather than refer to full regeneration.
2. The authors argue that bicarbonate formation in the absorber unit is not significant because "it does not require a bi-carbonate regeneration unit to recover CO₂" (response to question 7). The reviewer is still unclear on why this would not be an issue for the absorber, but it is for the electrolyzer, especially given that the same bicarbonate from the capture solution is then fed directly to the electrolyzer unit. If the authors mean to make an argument that the relative amounts or severity of bicarbonate formation are somehow different in the two scenarios (non-integrated and then integrated), they need to be much more explicit about this and provide numbers to back it up.
3. p. 3, line 73 – The authors cite an energy savings of 155-203 kJ/mol by avoiding regeneration. Are these numbers state of the art? Renfrew et al. (ACS Catal 2020, 10, 13058) claim 88 kJ/mol in more recent years, also see associated references therein.
4. p. 4, line 91 – "Unlike the sequential route, the captured CO₂ cannot be separated and recycled back to the electrolyser in the integrated route." It is not clear why this would be the case. The same CO₂ bound to amine can be recycled continuously, and eventually reacted.
5. Continuing, "If the capture medium cannot be fully recovered in the initial electrolysis step, it will cause an energy and capital penalty for the absorption or conversion step in the integrated electrolyser." Again, it is not clear why this would be the case, especially given that the authors revised their manuscript to use a lower lean loading of 0.3. Do the authors actually mean that the system is not operating between steady-state loading points? Otherwise, this statement with the ability to tolerate 0.3 minimum loading seem contradictory.
6. p. 7, line 182 – the authors invoke lower ionic conductivity of amine systems, but this seems somewhat arbitrary, because ionic conductivity can be readily increased (e.g. by addition of supporting electrolyte, etc.). The current limitations of amine systems should not be taken as fixed given the early stage nature of the field. In the following lines 184-185, the authors refer then to high voltage and energy requirements of early capture solutions; is this for the reason mentioned above? How does the assessment change if this perceived penalty is presumably addressed in coming years? The reviewer would argue that this feature simply hasn't been important enough to optimize yet given more basic issues with understanding reaction pathways and Faradaic efficiencies, but sees no real reason why this can't be addressed.
7. On p. 13, the authors mention a study where Cs⁺ and K⁺ were proposed to destabilize carbamate. This is not strictly true. K⁺ ions allow for formation of carbamic acid, which actually has a higher loading than carbamate, but cannot be considered to destabilize it.

Reviewer #2 (Remarks to the Author):

The authors developed sequential and integrated CO₂ capture and electrolysis model in order to compare energy consumption. This concept of CO₂ utilization has been emerged recently, so the topic

is of importance. However, this reviewer is still not convinced that the novelty of this manuscript (energy comparison only between the two processes without rigorous process simulations.) is enough to be published in this esteemed journal.

First, there are several recent papers that carried out techno-economic analysis and/or life cycle assessment as well as energy calculation of electrochemical conversion of CO₂ captured in absorbents. Thus, this reviewer doesn't find any difference or superiority of this manuscript than those in the literature. Here is an example (<https://doi.org/10.1016/j.apenergy.2021.117768>): "Life cycle and economic analysis of chemicals production via electrolytic (bi)carbonate and gaseous CO₂ conversion". This paper compared economic and environmental impacts of three processes: gas CO₂ conversion with PSA, gas CO₂ conversion with distillation, and captured CO₂ conversion for production of syngas or formate.

Second, the authors considered the role of the single-pass conversion and added a secondary recovery step to reach $X_{out}=0.3$ in the integrated process. According to Supplementary Information, this step consists of amine regeneration and gas CO₂ electrochemical conversion, which means that the integrated process is same for the sequential process except using the electrolysis of captured CO₂ as a pretreatment.

Third, the authors considered CO only and ignored other by-product, such as hydrogen. Some energy used in the electrolyzer generates hydrogen molecules and this is also a valuable chemical.

Fourth, type of energy is different from equipment to equipment. So, the authors should distinguish them in Fig. 3. Electricity is used for electrolysis, compression, and PSA, while heat is used for regeneration of amine and bicarbonate. One possible solution is to use energy cost (\$/mol CO₂), instead of energy (kJ/mol CO₂).

According to the points mentioned above I do not recommend to accept this manuscript.

Point-by-point response to the reviewers' comments

Reviewer #1:

This version of the manuscript is significantly improved and provides more nuanced discussion. The authors have taken care to carefully justify their reasoning and numbers used in many places, making it easier to evaluate the assumptions and findings. The current manuscript is more rigorous and could be considered for publication pending the ability to adequately address remaining yet still significant key issues below. Some of these are points that the reviewer continues to challenge and would like to see elaborated and better justified.

We express our sincere gratitude for the reviewer's time and efforts in reviewing our manuscript again. The comments have greatly enhanced the clarity and accuracy of our work for an interdisciplinary audience. We aim for this revision to satisfy the remaining concerns.

1. (Relatively minor comment) In the revised manuscript the authors have updated their definition of full regeneration to now refer to 0.3 loading. While this is commended and, the reviewer feels, quite appropriate operationally, the language may still be confusing to readers who may expect "full" to refer to 0 loading. The authors are recommended to polish this term, maybe to just use "lean loading state" consistently rather than refer to full regeneration.

We have corrected these terms throughout the main text.

On Page 11:

*...we assumed that the integrated electrolyser can recover the capture media to a **lean loading state** where it is directly recycled to the absorber (see Fig. 1a).*

On Page 12:

*To recover the CO₂-lean stream **to the lean loading state** after incomplete integrated electrolysis ...
... to regenerate the capture medium **to the lean loading state** and convert the rest of captured CO₂ to CO ...*

On Page 13:

*This analysis assumes the electrolyser can recover the capture medium to **the lean loading state**.*

On Page 16 – 17:

... and has a high single-pass conversion efficiency to achieve the CO₂-lean state of the amines.

2. The authors argue that bicarbonate formation in the absorber unit is not significant because “it does not require a (bi)carbonate regeneration unit to recover CO₂” (response to question 7). The reviewer is still unclear on why this would not be an issue for the absorber, but it is for the electrolyzer, especially given that the same bicarbonate from the capture solution is then fed directly to the electrolyzer unit. If the authors mean to make an argument that the relative amounts or severity of bicarbonate formation are somehow different in the two scenarios (non-integrated and then integrated), they need to be much more explicit about this and provide numbers to back it up.

The reviewer’s interpretation is correct. We are indeed making the argument that (bi)carbonate formation in the electrolysis step is different for the sequential and integrated route. Specifically, the degree of (bi)carbonate formation is much more severe in the non-integrated route than in the integrated route.

In the sequential route, (bi)carbonates are formed from the carbonation reaction between CO₂ gas in the electrolyser and the hydroxide ions produced from electrochemical reduction reactions (i.e., CO₂ electroreduction and hydrogen evolution reaction). The formed (bi)carbonates could migrate to the anolyte or precipitate at the electrode pores. These (bi)carbonates need to be recovered back to CO₂ so that most of the captured CO₂ can be converted to CO. Critically over 50% of the inputted CO₂ will be converted to (bi)carbonates.

In the integrated route, there is substantially less free CO₂ (i.e., both CO₂ (g) and CO₂ (aq)) than in the non-integrated route. Although hydroxide ions are still formed in the electroreduction process, these hydroxide ions will not react with excessive CO₂, because the majority of captured CO₂ is in the form of carbamate and minor bicarbonate.

To clarify this process, we included an additional explanation on Page 3:

In the CO₂ gas-fed electrolyser unit, CO₂ gas tends to form carbonate and bicarbonate ions (denoted as (bi)carbonates) by reacting with the hydroxide ions from electrochemical reduction (i.e., CO₂ reduction and hydrogen evolution reaction), as shown in R. 1-5. Usually, only less than 50% of CO₂ gas molecules consumed in the electrolyser contribute to CO production.¹⁷⁻¹⁹ The (bi)carbonates could either cross over the membrane²⁰ to the anolyte or precipitate at the cathode.²¹ The (bi)carbonates in the electrolyte can be regenerated back to CO₂ gas and hydroxide anolyte by reacting calcium hydroxide to form calcium carbonate precipitates. The precipitates will then be calcinated to release CO₂ and produce calcium oxide that will be hydrated to become calcium hydroxide in the final step.⁴

R. 1

On Page 4:

In the integrated route, the CO₂-rich amines, containing substantially less free CO₂ (i.e., both CO₂ (g) and CO₂ (aq))^{32,33}, are directly fed into the integrated electrolyzers. Although hydroxide ions are still produced from the electroreduction reactions, they will not react with free CO₂ in the integrated electrolysis, because the majority of the captured CO₂ molecules are in the form of carbamate and bicarbonate. As such, the integrated electrolysis inherently avoids CO₂ gas loss faced by the gas-fed electrolyzers.^{18,34-38}

Therefore, there is no need for the integrated route to include a (bi)carbonate regeneration unit. If fulfilled, the integrated process may save > 254 kJ mol_{CO₂}⁻¹ to recover the CO₂ and hydroxide from the (bi)carbonates.^{4,18}

3. p. 3, line 73 – The authors cite an energy savings of 155-203 kJ/mol by avoiding regeneration. Are these numbers state of the art? Renfrew et al. (ACS Catal 2020, 10, 13058) claim 88 kJ/mol in more recent years, also see associated references therein.

Thank you for the reference. We have updated the energy values in the main text and model to reflect the lower limits that have been reported.

In our modelling work, we maintain an average value of 179 kJ/mol_{CO₂} to reflect higher TRL proof of concept. We feel the work as presented also highlights how decreasing the absorber energy would influence the sequential vs integrated argumentation, so readers can appreciate how a reduction in energy would influence the system.

Changes made to the manuscript are highlighted below:

On Page 4:

In the amine-scrubbing cases, such a displacement could save 88 – 203 kJ mol_{CO₂}⁻¹ from amine regeneration^{22,25-29}

On Page 8 of the supplementary information:

The heat duty to separate CO_2 from amine solutions in the stripper is between $88 - 203 \text{ kJ mol}^{-1}$ if there is heat integration in the process.^{30,32,35,36} The baseline is assumed to be $179 \text{ kJ mol}_{\text{CO}_2}^{-1}$ to reflect a higher TRL proof of concept.

Fig. S6:

Fig. S6 Effects of operating conditions and solvent properties on the overall energy costs of the sequential and integrated route. The overall energy cost of the sequential route and integrated route as a function of **a**, single-pass conversion for gas-fed CO_2 electrolysis (note that the single-pass conversion is the ratio of total CO_2 consumed vs. the CO_2 feed), **b**, CO_2 loading in the effluent of the integrated electrolyser from capture media in the sequential route, **c**, energy required to regenerate amine-based capture medium with heat integration included, and **d**, the energy required to separate product from the effluent stream of gas-fed CO_2 electrolysers. The figures show baseline (green), pessimistic (grey), and optimistic (red) scenarios of the integrated route. The blue region is where the integrated route has energy advantages, while the orange region is vice versa.

4. p. 4, line 91 – “Unlike the sequential route, the captured CO₂ cannot be separated and recycled back to the electrolyser in the integrated route.” It is not clear why this would be the case. The same CO₂ bound to amine can be recycled continuously, and eventually reacted.

We agree that a recycle loop could in principle be implemented to continuously react the bound CO₂ to a low-loading state. In our statement, however, we are aiming to communicate that the integrated electrolyser itself may have mass transport limitations which prevent efficient operation (e.g. 90% selectivity at > 200 mA/cm²) at lower loadings. This is because the reactive species in the integrated electrolyzer must diffuse from the bulk electrolyte through a boundary layer of > 40 μm to reach the electrocatalytic surface (new Fig. S1a below). If the bulk concentration of reactive CO₂ species becomes too low, then the reactions occurring at the surface cannot be maintained. In the gas-fed electrolyzer, however, the diffusion thickness is on the order of nanometres (new Fig. S1b below) which enables substantial CO₂ access to the catalyst.

In brief, we agree our statement could be clearer, and we have greatly simplified our paragraph on page 4 to improve readership. The nuances of mass transport complexities in the integrated electrolyzer have then been fully shifted to “Single-pass conversion efficiency for the integrated electrolysis” section.

See added new Fig. S1:

Fig. S1 Comparison of the thickness of hydrodynamic boundary layers (δ_{BL}) at the cathode for (a) integrated electrolysis and (b) gas-fed electrolysis.

On Page 4:

Second, for the integrated process to fully remove the stripper process and regeneration energy consumption, the amine stream leaving the integrated electrolyser should be at the CO₂-lean loading state.

On Page 12:

If the electrolyser cannot recover the amine stream to a CO₂-lean state, unlike the sequential route, the excess captured CO₂ cannot be separated further from the effluent stream in the downstream separator and recycled back to the electrolyser for further CO₂ reduction in the integrated route. On one hand, if the lean CO₂ loading is unable to maintain a sufficient concentration gradient to drive the active species to the active sites in the electrolyser, it can degrade the CO selectivity and thus the energy efficiency. To maintain a high electrolysis efficiency, on the other hand, a higher CO₂ loading than the lean state can cause an increase in the size of the absorber to sustain the same absorption capacity. As such, a low single-pass CO₂ utilisation will cause energy and capital penalty for the absorption or conversion step in the integrated electrolyser.

In this case, the captured CO₂ in the effluent stream of the integrated electrolyser needs to be recovered to pure CO₂ gas from the regeneration unit and then fed into the gas-fed electrolyser for conversion. Due to the recycle loop in the gas-fed conversion step, the symbolic process can recover the CO₂-lean amine stream and convert the rest CO₂ into CO. Such function cannot be easily achieved only by the integrated electrolysis process, which is unable to maintain a high concentration of active species due to the absence of a downstream separation process to recover the CO₂-rich stream.

5. Continuing, “If the capture medium cannot be fully recovered in the initial electrolysis step, it will cause an energy and capital penalty for the absorption or conversion step in the integrated electrolyser.” Again, it is not clear why this would be the case, especially given that the authors revised their manuscript to use a lower lean loading of 0.3. Do the authors actually mean that the system is not operating between steady-state loading points? Otherwise, this statement with the ability to tolerate 0.3 minimum loading seem contradictory.

As discussed in the above comment we have changed the paragraph to be much clearer. Our original statement was meant as an initial discussion point for the work presented in Figure 4, but we have now fully shifted that discussion away from the introduction to avoid confusion.

In brief, our primary results (Figures 1, 2 and 3) in the manuscript compare the sequential and integrated routes assuming there are no constraints that would prevent the integrated electrolyzer from fully replacing the stripper unit (i.e., the outlet is at a lean loading of 0.3). The work in Figure 4 and the section *Single-pass conversion efficiency for the integrated electrolysis* then discusses the implications that mass transport limitations may have on achieving this assumption in practice.

Accordingly, we have revised our text on Page 4:

Second, for the integrated process to fully remove the stripper process and regeneration energy consumption, the amine stream leaving the integrated electrolyser should be at the CO₂-lean loading state.

On Page 7:

These CO₂ species then need to diffuse to the negatively charged electrode through a thick hydrodynamic boundary layer usually > 40 μm, similar to a CO₂-fed aqueous H-cell electrolyser.^{57,58} Such a thick boundary layer is anticipated to limit the mass transfer rate of the active species to the electrode surface and thus degrade CO Faradaic efficiency at industrially relevant current densities. (see Fig. S1) Therefore, we could foresee a scenario where the integrated electrolyser is unable to achieve a CO₂-lean state via a single pass.

On Page 12:

Due to the thick boundary layer existing in the integrated electrolyser, the active species near a catalyst surface will then be depleted even if the bulk concentration remains high.

If the electrolyser cannot recover the amine stream to a CO₂-lean state, unlike the sequential route, the excess captured CO₂ cannot be separated further from the effluent stream in the downstream separator and recycled back to the electrolyser for further CO₂ reduction in the integrated route. On one hand, if the lean CO₂ loading is unable to maintain a sufficient concentration gradient to drive the active species to the active sites in the electrolyser, it can degrade the CO selectivity and thus the energy efficiency. To maintain a high electrolysis efficiency, on the other hand, a higher CO₂ loading than the lean state can cause an increase in the size of the absorber to sustain the same absorption capacity. As such, a low single-pass CO₂ utilisation will cause energy and capital penalty for the absorption or conversion step in the integrated electrolyser.

6. p. 7, line 182 – the authors invoke lower ionic conductivity of amine systems, but this seems somewhat arbitrary, because ionic conductivity can be readily increased (e.g. by addition of supporting electrolyte, etc.). The current limitations of amine systems should not be taken as fixed given the early stage nature of the field. In the following lines 184-185, the authors refer then to high voltage and energy requirements of early capture solutions; is this for the reason mentioned above? How does the assessment change if this perceived penalty is presumably addressed in coming years? The reviewer would argue that this feature simply hasn't been important enough to optimize yet given more basic issues with understanding reaction pathways and Faradaic efficiencies, but sees no real reason why this can't be addressed.

Thank you for the Reviewer's comment. Accordingly, we have added new discussions on the ionic conductivity improvement in the main text. The assumptions of higher ionic conductivities have then also

been used in a new Figure 2a. See below for the old (Figure R1 left) and new (Figure R1 right) panels where the predicted cell voltage of the referenced works (red circles) have been decreased due to the increased ionic conductivities. This is most relevant for the larger current density data points.

Figure R1: Changes to the predicted cell voltages of the previous experimental with the assumption of high ionic conductivities.

On Page 7:

The higher energy requirement of the integrated system is a result of the lower CO selectivity than the gas-fed systems.

Taking Lee et al.'s result as an example, the estimated cell voltage is 3 V to achieve 100 mA cm^{-2} assuming the amine solution has the same ionic conductivity of 1 M KOH aqueous solution (21.5 S m^{-1} for 1 M KOH solution⁵⁹).²³ The amine aqueous solution has a lower ionic conductivity than inorganic electrolyte (i.e. 3.7 S m^{-1} for 5 M monoethanolamine solutions with about $0.4 \text{ mol}_{\text{CO}_2} \text{ mol}_{\text{amine}}^{-1}$ ⁶⁰ as compared to 21.5 S m^{-1} for 1 M KOH solution⁵⁹). The ionic conductivity of the capture media can be effectively improved by including inorganic salts, such as K_2SO_4 and KCl .^{23,61} As a result, the ohmic loss from the capture solvent can be significantly reduced, which is shown in Fig. S2.

Further, the halide ions can serve as inhibitors to prevent oxidative degradation of amines,^{62,63} and the alkali cations are effective in promoting CO_2 electrochemical conversion.^{23,61,64} Buvik et al.⁶³ also reported that the NaCl and KI salts show negligible impacts on the CO_2 capture capacity of the 30 wt% monoethanolamine solution. Nevertheless, further research efforts are needed to investigate the impacts of other inorganic salts on the properties of the capture media and the CO_2 absorption performance.

Fig. 2a:

Fig. 1 Energy cost to convert CO_2 to CO for gas-fed CO_2 electrolyser and direct CO_2 electrochemical upgrade from capture medium. **a**, The energy cost to convert CO_2 to CO as a function of CO Faradaic efficiency with recently reported values for two different CO_2 electrolysers. Detailed data and references are summarized in Tables S2 and S3. The bubble size represents the magnitude of current densities for these cells as indicated in the legend. The insets illustrate the operating conditions of these two cells. **b**, Impacts of CO Faradaic efficiency and cell voltages on the energy cost of the CO_2 electrolysers. The solid lines indicate the Faradaic Efficiency vs. Cell voltage trends at certain energy requirements as indicated inline.

Fig. S2:

Fig. S2 Relation between current densities and the potentials (without cathode potential) for the integrated CO_2 electrolyser with (a) 30 wt% monoethanolamine solution as the catholyte and (b) 30 wt% monoethanolamine solution with inorganic salts as the catholyte that has the same ionic conductivity (21.5 S m^{-1}) for the 1 M KOH.

7. On p. 13, the authors mention a study where Cs⁺ and K⁺ were proposed to destabilize carbamate. This is not strictly true. K⁺ ions allow for formation of carbamic acid, which actually has a higher loading than carbamate, but cannot be considered to destabilize it.

We have deleted the associated sentences on Page 14-15 in the main text to improve the accuracy. Thanks for the correction.

Reviewer #2:

The authors developed sequential and integrated CO₂ capture and electrolysis model in order to compare energy consumption. This concept of CO₂ utilization has emerged recently, so the topic is of importance. However, this reviewer is still not convinced that the novelty of this manuscript (energy comparison only between the two processes without rigorous process simulations.) is enough to be published in this esteemed journal.

First, we would like to appreciate the Reviewer's additional time for reviewing our manuscript and providing detailed feedback. We also appreciate the reviewer's recognition of the importance of this topic. We have revised the manuscript accordingly to improve the quality and clarity of our work, which we hope allows for reconsideration of its novelty in the context of amine-based capture with direct conversion.

First, there are several recent papers that carried out techno-economic analysis and/or life cycle assessment as well as energy calculation of electrochemical conversion of CO₂ captured in absorbents. Thus, this reviewer doesn't find any difference or superiority of this manuscript than those in the literature. Here is an example (<https://doi.org/10.1016/j.apenergy.2021.117768>): "Life cycle and economic analysis of chemicals production via electrolytic (bi)carbonate and gaseous CO₂ conversion". This paper compared economic and environmental impacts of three processes: gas CO₂ conversion with PSA, gas CO₂ conversion with distillation, and captured CO₂ conversion for production of syngas or formate.

We have carefully read the suggested paper and noticed that our work has a very different scope and application. Our work's primary novelties focused on the requirements and detailed operation of an integrated electrolyzer as described below, which have not been addressed in existing works. Further, our conclusions provide simple and key performance targets for researchers in the experimental electrolysis research community, while challenging the often simplified assumptions regarding the benefits of integration.

In brief, our key novelties are highlighted below:

- Our findings show that the integration route could save up to 44% energy (mainly thermal energy) as compared to the non-integrated route when using amine-based CO₂ capture processes.
- As the electrolyzer energy cost is dominant, however, the integrated electrolyser must achieve similar CO Faradaic efficiency (90%) and cell voltages (3 V) to the gas-fed electrolyser to realize energy gains from avoiding the CO₂ stripper.
- Despite the natural tendency of the field to believe the integrated electrolysis can recover amine to CO₂-lean state, we express the fundamental and applied challenges (e.g., mass transport limitations

and potentially low single-pass conversion efficiency due to > 40 μm hydrodynamic boundary layer) that have to be addressed before making the integrated electrolysis viable.

Accordingly, we revised our abstract and conclusions in the main text to further clarify the novelty of our work:

On Page 1:

Our high-level energy analyses unveil that an integrated electrolyser must show similar performance to the gas-fed electrolyser and high single-pass conversion efficiency to ensure an energy benefit of up to 44% versus the sequential route.

On Page 16:

Lastly, a directly coupled CO₂ capture and electrochemical conversion could potentially save close to 44% energy consumption and 21% energy cost versus a sequential process based on the state-of-the-art gas-fed CO₂ electrolysers, if the integrated electrolysis performs similarly to the gas-fed electrolysis (3 V and 90% CO Faraday efficiency) and has a high single-pass conversion efficiency to achieve the CO₂-lean state of the amines.

Second, the authors considered the role of the single-pass conversion and added a secondary recovery step to reach $X_{\text{out}}=0.3$ in the integrated process. According to Supplementary Information, this step consists of amine regeneration and gas CO₂ electrochemical conversion, which means that the integrated process is same for the sequential process except using the electrolysis of captured CO₂ as a pretreatment.

In this separate section entitled *Single-pass conversion efficiency for the integrated electrolysis*, we highlighted the importance of achieving a high single-pass CO₂ conversion efficiency when designing an efficient integrated electrolyser. In fact, this is a key novelty not presented in other works which incorrectly assume that liquid-fed electrolysis units can reach similar current densities to state-of-the-art gas-fed systems.

In brief, our results generated new insights that have been ignored by the field: the recovery of the amines to the CO₂-lean state in the integrated electrolyser cannot be taken for granted. The integrated electrolyser could face mass transport limitation due to the thick (> 40 μm) hydrodynamic boundary layer at the cathode. It means that the active species could be depleted at the electrode surface even if their concentration is high in the bulk.

Therefore, we have considered the steps and possibilities to complete the loop of integrated CO₂ capture and electrolysis, such as incorporation of a downstream processes including amine regeneration, CO₂

electrolysis, product separation, and potential CO₂ recovery from (bi)carbonates. We also used a separate section to assess the energy penalty for an incomplete recovery of CO₂-lean amines.

We have revised our main text to improve the clarity:

On Page 4:

Second, for the integrated process to fully remove the stripper process and regeneration energy consumption, the amine stream leaving the integrated electrolyser should be at the CO₂-lean loading state.

On Page 7:

*These CO₂ species then need to diffuse to the negatively charged electrode through a **thick** hydrodynamic boundary layer usually > 40 μm, similar to a CO₂-fed aqueous H-cell electrolyser.^{57,58} Such a thick boundary layer is anticipated to limit the mass transfer rate of the active species to the electrode surface and thus degrade CO Faradaic efficiency at industrially relevant current densities. (see Fig. S1) Therefore, we could foresee a scenario where the integrated electrolyser is unable to achieve a CO₂-lean state via a single pass.*

New Fig. S1:

Fig. S1 Comparison of the thickness of hydrodynamic boundary layers (δ_{BL}) at the cathode for (a) integrated electrolysis and (b) gas-fed electrolysis.

On Page 12:

Due to the thick boundary layer existing in the integrated electrolyser, the active species near a catalyst surface will then be depleted even if the bulk concentration remains high.

If the electrolyser cannot recover the amine stream to a CO₂-lean state, unlike the sequential route, the excess captured CO₂ cannot be separated further from the effluent stream in the downstream separator and recycled back to the electrolyser for further CO₂ reduction in the integrated route. On one hand, if the lean CO₂ loading is unable to maintain a sufficient concentration gradient to drive the active species to the active sites in the electrolyser, it can degrade the CO selectivity and thus the energy efficiency. To maintain a high electrolysis efficiency, on the other hand, a higher CO₂ loading than the lean state can cause an increase in the size of the absorber to sustain the same absorption capacity. As such, a low single-pass CO₂ utilisation will cause energy and capital penalty for the absorption or conversion step in the integrated electrolyser.

In this case, the captured CO₂ in the effluent stream of the integrated electrolyser needs to be recovered to pure CO₂ gas from the regeneration unit and then fed into the gas-fed electrolyser for conversion. Due to the recycle loop in the gas-fed conversion step, the symbolic process can recover the CO₂-lean amine stream and convert the rest CO₂ into CO.

Third, the authors considered CO only and ignored other by-product, such as hydrogen. Some energy used in the electrolyzer generates hydrogen molecules and this is also a valuable chemical.

We have assumed in our model that hydrogen contributes to the rest of Faradaic efficiency, and the energy associated with by-product hydrogen production has been considered in the analysis. We consider H₂ as an undesired product, however, as it is cheaper to produce hydrogen using water electrolysis than using CO₂ electrolysis. Further, the majority of downstream CO reactions include H₂. Therefore, this work is mainly focused on the energy required to capture and conversion of CO₂, as these factors will determine the design of such an integrated electrolyzer unit.

Accordingly, we have revised the main text.

On Page 5:

Hydrogen is usually evolved as a side product together with CO₂ conversion, and CO product is mainly used together with hydrogen as the feedstock for downstream chemical manufacturing.⁴⁸ As such, this study is mainly focused on the energy required for CO₂ abatement over the CO₂ capture and conversion process.

Fourth, type of energy is different from equipment to equipment. So, the authors should distinguish them in Fig. 3. Electricity is used for electrolysis, compression, and PSA, while heat is used for regeneration of amine and bicarbonate. One possible solution is to use energy cost (\$/mol CO₂), instead of energy (kJ/mol CO₂).

According to the Reviewer's comments, we have revised our models, figures, and main text to account for the differences in thermal and electrical energy input. We have added two separate figure panels to Figure 3 to show energy breakdown as well as cost per ton of converted CO₂. We thank the reviewer for this input as it provides an additional perspective to analyze these systems.

See below for changes made to the manuscript and supplementary information:

Fig. 3 Scenario analysis of (a) overall energy consumption, (b) thermal energy and electricity consumption, and (c) energy cost for sequential and integrated routes. In the sequential route, the CO₂

electrolyser includes state-of-the-art gas-fed electrolysers that show 50% CO₂ utilisation or future scenarios with 100% CO₂ utilisation. The optimistic, baseline and pessimistic electrolysis cases for the integrated routes are compared against the sequential route.

On Page 8:

We compared the sequential and integrated routes in terms of total energy, thermal energy and electricity, and energy cost.

On Page 9:

Considering the higher cost of electricity than heat, the integrated route in the baseline is in fact inferior to the sequential route due to its high electrical energy consumption (see Fig. 3b and c).

On Page 10 – 11:

In this scenario, the integrated route can save up to 44% of total energy due to a low cell voltage, high CO Faradaic efficiency, and no thermal energy associated with regeneration of amines (179 kJ mol_{CO₂}⁻¹) and (bi)carbonate (254 kJ mol_{CO₂}⁻¹), and electricity associated with CO₂ compression (17 kJ mol_{CO₂}⁻¹), and product purification (51 kJ mol_{CO₂}⁻¹). (Fig. 3b) The integrated route could save 22% energy cost over the sequential route.

On Page 11:

Here the integrated optimistic case then only maintains a maximum overall energy advantage of 26% and energy cost benefit of 11%.

REVIEWER COMMENTS

Reviewer #1 (Remarks to the Author):

My more minor comments have been addressed in this version. However I still have some doubts about the scientific tone of the article, which seems to put forward some ideas as a done deal when experimentalists have barely started working on this topic. I would encourage the authors to make sure they are sticking to clearly stated and justified assumptions and impartial analyses and avoid speculation where it's not necessary, regarding the points noted below. The manuscript does otherwise have merits and I would be willing to recommend publication should these last issues be given final consideration.

1. I still have some doubts about the argument put forward regarding CO₂ loading state. In the revision the authors write that "If the electrolyser cannot recover the amine stream to a CO₂-lean state, unlike the sequential route, the excess captured CO₂ cannot be separated further from the effluent stream in the downstream separator and recycled back to the electrolyser for further CO₂ reduction in the integrated route." The reviewer feels this argument is still specious, especially given that loading studies have hardly been done in the electrochemical processes. Why can't the CO₂-excess loading solution simply be exposed to CO₂ again to reload the solution? This would imply a higher lean-loading point and may result in decreased absorption rates, but is not necessarily a non-starter. The argument about a thick hydrodynamic boundary layer might be true, but there are types of reactors that can try to thin this layer, and it feels premature to rule on this point given that custom reactors have not yet been designed for the integrated process. In practice, some of these are engineering considerations, so it feels unfair to penalize the integrated systems based on possible issues that might arise, yet about which essentially no experimental research has actually been done.
2. "Second, for the integrated process to fully remove the stripper process and regeneration energy consumption, the amine stream leaving the integrated electrolyser should be at the CO₂-lean loading state." This statement appears to me to simply be saying that some kind of steady-state condition at the outlet is needed, which is certainly not surprising for an appropriately designed process. Yet it is put forward as a drawback.
3. Are the authors totally sure that produced hydroxide anions won't react with CO₂ bound in the form of carbamate? Carbamates hydrolyze. There are complex equilibria going on there that seem not simplistically brushed away.

Point-by-point responses to the reviewer's comments

My more minor comments have been addressed in this version. However I still have some doubts about the scientific tone of the article, which seems to put forward some ideas as a done deal when experimentalists have barely started working on this topic. I would encourage the authors to make sure they are sticking to clearly stated and justified assumptions and impartial analyses and avoid speculation where it's not necessary, regarding the points noted below. The manuscript does otherwise have merits and I would be willing to recommend publication should these last issues be given final consideration.

Thank you again for the reviewer's time and efforts in reviewing our manuscript. We aim for this revision to satisfy the remaining concerns. In particular, we have done our best to improve the clarity and objectivity of the below concerns.

1. I still have some doubts about the argument put forward regarding CO₂ loading state. In the revision the authors write that "If the electrolyser cannot recover the amine stream to a CO₂-lean state, unlike the sequential route, the excess captured CO₂ cannot be separated further from the effluent stream in the downstream separator and recycled back to the electrolyser for further CO₂ reduction in the integrated route."

1.1 The reviewer feels this argument is still specious, especially given that loading studies have hardly been done in the electrochemical processes. Why can't the CO₂-excess loading solution simply be exposed to CO₂ again to reload the solution? This would imply a higher lean-loading point and may result in decreased absorption rates, but is not necessarily a non-starter.

1.2 The argument about a thick hydrodynamic boundary layer might be true, but there are types of reactors that can try to thin this layer, and it feels premature to rule on this point given that custom reactors have not yet been designed for the integrated process. In practice, some of these are engineering considerations, so it feels unfair to penalize the integrated systems based on possible issues that might arise, yet about which essentially no experimental research has actually been done.

Comment 1.1: We agree with the reviewer that increasing the lean loading point is a viable technological and economical option and have expanded this discussion in the text and an SI figure for illustration (see below). We note that this option is at a higher TRL than the alternate option of requiring higher conversion in the integrated electrolyzer. Our discussion text now discusses both options more objectively, and more clearly states that our analysis assesses the second option which is currently unexplored but is not necessarily preferential to operating with a higher lean-loading state and larger absorber.

On Page 11 – 12:

In the analyses above, we assumed that the integrated electrolyser could recover the capture media to a lean loading state where it is directly recycled to the absorber (see Fig. 1b). If the electrolyser is unable to achieve the proposed lean state of $0.3 \text{ mol}_{\text{CO}_2} \text{ mol}_{\text{amine}}^{-1}$, the high CO₂ loading ($X > 0.3$) in the lean amine stream will decrease the CO₂ absorption rate in the absorber unit. To maintain the overall CO₂ capture

and conversion capacity of the process, adjustments to the process in Fig. 1b would then be needed. Here we discuss two possibilities, both of which will incur either additional capital or energy costs for the process.

One possible adjustment to account for lower conversions in the integrated electrolyzer is to increase the size of the absorber unit (Fig. S4). A smaller difference between the low and high CO₂ loading states will then be present and a larger absorber allows for the same CO₂ capture capacity. Previous reports analysing the impacts on absorber size and capture costs of higher lean loading states indicate that an increase in lean loading from 0.3 to > 0.4 mol_{CO₂} mol_{amine}⁻¹ would require 20-38% more capture costs.^{63,64} With the electrolyser unit dominating the energy costs, however, these increased capture costs would be less substantial when considering the complete process. This option is also at a high technology readiness level.

A second option to maintain CO₂ capture and conversion capacity would be to add a secondary step after the integrated electrolyzer, which is a smaller version of the stripper and gas-fed electrolyzer unit from the sequential process (Fig. 4a). The energy implications of this option have yet to be explored in literature and will be examined within this section. In essence, this analysis examines the role of the single-pass conversion of the integrated electrolyser.

In the model, we included a symbolic process (including amine regeneration, gas-fed CO₂ electrolysis, product separation, and (bi)carbonate regeneration, shown in Fig. 4a) to regenerate the capture medium to the lean loading state and convert the rest of captured CO₂ to CO. In this case, the captured CO₂ in the effluent stream of the integrated electrolyser needs to be recovered to pure CO₂ gas from the regeneration unit and then fed into the gas-fed electrolyser for conversion.

On Page 15 of the supplementary information, we included a new Figure S4:

Fig. S 1 A schematic illustration of the integrated route that requires a sizeable absorber to account for the incomplete conversion in the integrated electrolysis system.

Comment 1.2: We also agree that the conclusions on the boundary layer may be premature. We have then changed the discussion to note that mass transport in the integrated electrolyzer is an important consideration for future research, but we no longer indicate that it may be prohibitive or unlikely to be resolved.

Changes were then made to the manuscript on pages 7, 12, 14 and 16 as highlighted below:

On Page 7, we deleted the related statement, and now it becomes:

*In contrast to the gas-fed system above, reported electroreduction of captured CO₂ in monoethanolamine solutions presently has a higher energy requirement at low current densities (**Error! Reference source not found.**a and Table S2). The higher energy requirement of the integrated system is a result of the lower CO selectivity than the gas-fed systems. **With further research efforts, these metrics are expected to improve.***

~~*The catalytically active species for CO₂ conversion in the amine solutions remain unclear but should be related to the free CO₂ dissolved in the solution, carbamate and bicarbonate ions, partly or all present in a CO₂-rich amine stream. These CO₂ species then need to diffuse to the negatively charged electrode through a thick hydrodynamic boundary layer usually > 40 μm, similar to a CO₂-fed aqueous H cell electrolyser.^{57,58} Such a thick boundary layer is anticipated to limit the mass transfer rate of the active species to the electrode surface and thus degrade CO Faradaic efficiency at industrially relevant current densities. (see Fig. S1) Therefore, we could foresee a scenario where the integrated electrolyser is unable to achieve a CO₂-lean state via a single pass.*~~

On Page 12, we also removed the following paragraph:

~~*If the electrolyser cannot recover the amine stream to a CO₂-lean state, unlike the sequential route, the excess captured CO₂ cannot be separated further from the effluent stream in the downstream separator and recycled back to the electrolyser for further CO₂ reduction in the integrated route. On one hand, if the lean CO₂ loading is unable to maintain a sufficient concentration gradient to drive the active species to the active sites in the electrolyser, it can degrade the CO selectivity and thus the energy efficiency. To maintain a high electrolysis efficiency, on the other hand, a higher CO₂ loading than the lean state can cause an increase in the size of the absorber to sustain the same absorption capacity. As such, a low single pass CO₂ utilisation will cause energy and capital penalty for the absorption or conversion step in the integrated electrolyser.*~~

On Page 15, we clarify the assumption of our argument:

Additionally, the active species need to diffuse to the negatively charged electrode through a thick hydrodynamic boundary layer usually $> 40 \mu\text{m}$, if the integrated reactor configuration is similar to a CO_2 -fed aqueous H-cell electrolyser (see Fig. S7 for a comparison of aqueous versus gas-fed mass transport in CO_2 electrolysis).^{68,69} Efforts to improve integrated conversion at elevated current densities should then take such transport into consideration when designing such systems.

On Page 16:

We anticipate a significant improvement in CO_2 conversion rates ($> 200 \text{ mA cm}^{-2}$) by implementing new electrode structures such as hydrophilic 3D structured flow-through electrodes and optimised capture media.^{75,76} The required diffusion distances of active species to achieve industrially applicable current densities are highly dependent on the concentrations and diffusion coefficients of the active species.⁷⁷ Therefore, understanding the primary active species and tailoring the local reaction environment could be effective in enhancing the CO_2 conversion rate in the integrated electrolysers.

Using metallic porous flow through electrodes is expected to achieve a high rate of CO_2 conversion by maximizing the electrochemical surface area, reducing the thickness of the boundary layer, and accelerating the detachment of gas products.

2. “Second, for the integrated process to fully remove the stripper process and regeneration energy consumption, the amine stream leaving the integrated electrolyser should be at the CO_2 -lean loading state. “This statement appears to me to simply be saying that some kind of steady-state condition at the outlet is needed, which is certainly not surprising for an appropriately designed process. Yet it is put forward as a drawback.

We agree that the statement as phrased is unclear. We now have greatly simplified our paragraph on Page 4 to enhance the readership.

On Page 4:

However, there should be additional requirements for integrated electrolysis to be beneficial and replace amine regeneration and CO_2 compression in the sequential process. For instance, the integrated electrochemical conversion step needs to show at least similar performance metrics (cell voltage, Faradaic efficiency, and current densities) as the gas-fed electrolysers in the sequential process. Otherwise, energy gains for the overall process may be offset by the increased electrolyser energy requirements. Therefore, it is not straightforward to compare the energy benefits of an integrated process, thus warranting a more detailed analysis to help determine the upper limits of this new research direction.

3. Are the authors totally sure that produced hydroxide anions won't react with CO₂ bound in the form of carbamate? Carbamates hydrolyze. There are complex equilibria going on there that seem not simplistically brushed away.

We agree that the reaction equilibria inside the integrated electrolysis system are complex. We have revised our paragraph to provide greater scope of known and unknown reactions. The changes made are highlighted below.

What are the pathways for the regeneration of the capture media?

Complex homogenous equilibrium reactions often take place in the CO₂-capture medium system. In the sequential route, heating is required to drive the reactions towards the recovery of capture media and CO₂. Whereas the integrated route, as shown in Fig. 1b, uses electrochemical reactions to regenerate the capture medium via reduction of absorbed CO₂ and chemical-induced equilibria shift to the original states of the capture medium (see an example in Fig. 5a). Therefore, understanding the reaction equilibria under CO₂ electroreduction conditions is vital to the identification of chemical pathways to recover capture media inside the integrated electrolyser.

Similar to the gas-fed CO₂ electroreduction, hydroxide ions should also be produced at the catalyst surface as a by-product of water reduction and increase the pH locally around the electrode.⁷⁰ A prior report⁷¹ has shown that the addition of a strong base (e.g., sodium hydroxide) to the CO₂-amine system could result in the formation of free amines and carbonate at the end equivalent points. As such, we could anticipate the formation of carbonate ions close to the electrode surface from the reactions between the hydroxide ions and unreacted CO₂ species. These carbonate ions could either reverse back to carbamate, free CO₂, or bicarbonate by reacting with the protons from the membrane^{71,72} or stay as carbonate if additional cations are introduced into the cathode channel. The latter situation may cause operational issues for the integrated route such as inefficient CO₂ conversion, alteration of solvent chemistry, and potential carbonate salt precipitation from the solvent. Hence a dedicated control and balance of ions within the electrolyser also become critical in achieving an efficient amine recovery when using electrochemical CO₂ reduction as a regeneration step.

REVIEWERS' COMMENTS

Reviewer #1 (Remarks to the Author):

The changes are satisfactory and I am pleased to recommend publication of the article at this time.